# DATA-DRIVEN OFFLINE OPTIMIZATION FOR ARCHITECTING HARDWARE ACCELERATORS

**Aviral Kumar**[†,*]   **Amir Yazdanbakhsh**[†]
**Milad Hashemi**   **Kevin Swersky**   **Sergey Levine**[*]

Google Research   [*] UC Berkeley   ([†] Equal Contribution)
aviralk@berkeley.edu, ayazdan@google.com

## ABSTRACT

To attain higher efficiency, the industry has gradually reformed towards application-specific hardware accelerators. While such a paradigm shift is already starting to show promising results, designers need to spend considerable manual effort and perform large number of time-consuming simulations to find accelerators that can accelerate multiple target applications while obeying design constraints. Moreover, such a "simulation-driven" approach must be re-run from scratch every time the set of target applications or design constraints change. An alternative paradigm is to use a "data-driven", offline approach that utilizes logged simulation data, to architect hardware accelerators, without needing any form of simulations. Such an approach not only alleviates the need to run time-consuming simulation, but also enables data reuse and applies even when set of target applications changes. In this paper, we develop such a data-driven offline optimization method for designing hardware accelerators, dubbed PRIME, that enjoys all of these properties. Our approach learns a conservative, robust estimate of the desired cost function, utilizes infeasible points and optimizes the design against this estimate without any additional simulator queries during optimization. PRIME architects accelerators—tailored towards both single- and multi-applications—improving performance upon stat-of-the-art simulation-driven methods by about $1.54\times$ and $1.20\times$, while considerably reducing the required total simulation time by 93% and 99%, respectively. In addition, PRIME also architects effective accelerators for unseen applications in a zero-shot setting, outperforming simulation-based methods by $1.26\times$.

## 1 INTRODUCTION

The death of Moore's Law [11] and its spiraling effect on the semiconductor industry have driven the growth of specialized hardware accelerators. These specialized accelerators are tailored to specific applications [64, 47, 41, 53]. To design specialized accelerators, designers first spend considerable amounts of time developing simulators that closely model the real accelerator performance, and then optimize the accelerator using the simulator. While such simulators can automate accelerator design, this requires a large number of simulator queries for each new design, both in terms of simulation time and compute requirements, and this cost increases with the size of the design space [65, 53, 25]. Moreover, most of the accelerators in the design space are typically infeasible [25, 64] because of build errors in silicon or compilation/mapping failures. When the target applications change or a new application is added, the complete simulation-driven procedure is generally repeated. To make such approaches efficient and practically viable, designers typically "bake-in" constraints or otherwise narrow the search space, but such constraints can leave out high-performing solutions [9, 44, 7].

An alternate approach, proposed in this work, is to devise a *data-driven* optimization method that only utilizes a database of previously tested accelerator designs, annotated with measured performance metrics, to produce new optimized designs *without* additional active queries to an explicit silicon or a cycle-accurate simulator. Such a data-driven approach provides three key benefits: **(1)** it significantly shortens the recurring cost of running large-scale simulation sweeps, **(2)** it alleviates the need to explicitly bake in domain knowledge or search space pruning, and **(3)** it enables data re-use by empowering the designer to optimize accelerators for new unseen applications, by the virtue of effective generalization. While data-driven approaches have shown promising results in

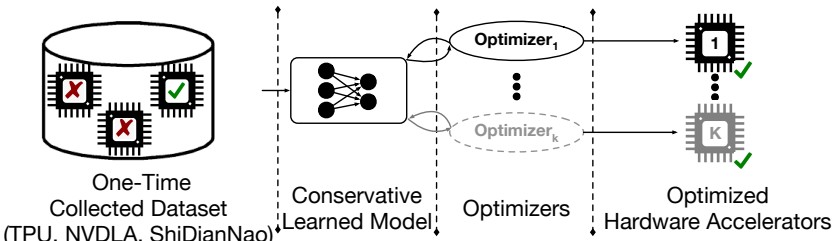

**Figure 1: Overview of PRIME.** We use a one-time collected dataset of prior accelerator designs, including TPU-style [65], NVDLA-style [42], and ShiDianNao-style [10] accelerators to train a conservative surrogate model, which is used to design accelerators to meet desired goals and constraints.

biology [14, 5, 57], using offline optimization methods to design accelerators has been challenging primarily due to the abundance of infeasible design points [64, 25].

The key contribution of this paper is a data-driven approach, PRIME , to automatically architect high-performing application-specific accelerators by using only previously collected offline data. PRIME learns a robust surrogate model of the task objective function from an existing offline dataset, and finds high-performing application-specific accelerators by optimizing the architectural parameters against this learned surrogate function, as shown in Figure 1. While naïvely learned surrogate functions usually produces poor-performing, out-of-distribution designs that appear quite optimistic under the learned surrogate [35, 5, 57]. The robust surrogate in PRIME is explicitly trained to prevent overestimation on "adversarial" designs that would be found during optimization. Furthermore, in contrast to prior works that discard infeasible points [25, 57], our proposed method instead incorporates infeasible points when learning the conservative surrogate by treating them as additional negative samples. During evaluation, PRIME optimizes the learned conservative surrogate.

Our results show that PRIME architects hardware accelerators that improve over the best design in the training dataset, on average, by 2.46× (up to 6.7×) when specializing for a single application. In this case, PRIME also improves over the best conventional simulator-driven optimization methods by 1.54× (up to 6.6×). These performance improvements are obtained while reducing the total simulation time to merely 7% and 1% of that of the simulator-driven methods for single-task and multi-task optimization, respectively. More importantly, a contextual version of PRIME can design accelerators that are *jointly optimal* for a set of *nine* applications without requiring any additional domain information. In this challenging setting, PRIME improves over simulator-driven methods, which tend to scale poorly as more applications are added, by 1.38×. Finally, we show that the surrogates trained with PRIME on a set of training applications can be readily used to obtain accelerators for *unseen* target applications, without any retraining on the new application. Even in this *zero-shot* optimization scenario, PRIME outperforms simulator-based methods that require re-training and active simulation queries by up to 1.67×. In summary, PRIME allows us to effectively address the shortcomings of simulation-driven approaches, (1) significantly reduces the simulation time, (2) enables data reuse and enjoys generalization properties, and (3) does not require domain-specific engineering or search space pruning.

## 2 BACKGROUND ON HARDWARE ACCELERATORS

The goal of specialized hardware accelerators—Google TPUs [29, 23], Nvidia GPUs [43], Graph-Core [21]—is to improve the performance of specific applications, such as machine learning models. To design such accelerators, architects typically create a parameterized design and sweep over parameters using simulation.

**Target hardware accelerators.** Our primary evaluation uses an industry-grade and highly parameterized template-based accelerator following prior work [65]. This template enables architects to determine the organization of various components, such as compute units, memory cells, memory, etc., by searching for these configurations in a discrete design space. Some ML applications may have large memory requirements (e.g., large language models [6]) demanding sufficient on-chip memory resources, while others may benefit from more compute blocks. The hardware design workflow directly selects the values of these parameters. In addition to this accelerator and to further show the generality of our method to other accelerator design problems, we evaluate two distinct dataflow accelerators with different search spaces, namely NVDLA-style [42] and ShiDianNao-style [10] from Kao et al. [30] (See Section 6 and Appendix C for a detailed discussion; See Table 6 for results).

**Table 1:** The accelerator design space parameters for the primary accelerator search space targeted in this work. The maximum possible number of accelerator designs (including feasible and infeasible designs) is 452,760,000. PRIME only uses a small randomly sampled subset of the search space.

| Accelerator Parameter | # Discrete Values | Accelerator Parameter | # Discrete Values |
|---|---|---|---|
| # of PEs-X | 10 | # of PEs-Y | 10 |
| PE Memory | 7 | # of Cores | 7 |
| Core Memory | 11 | # of Compute Lanes | 10 |
| Instruction Memory | 4 | Parameter Memory | 5 |
| Activation Memory | 7 | DRAM Bandwidth | 6 |

**How does an accelerator work?** We briefly explain the computation flow on our template-based accelerators (Figure 2) and refer the readers to Appendix C for details on other accelerators. This template-based accelerator is a 2D array of processing elements (PEs). Each PE is capable of performing matrix multiplications in a single instruction multiple data (SIMD) paradigm [20]. A controller orchestrates the data transfer (both activations and model parameters) between off-chip DRAM memory and the on-chip buffers and also reads in and manages

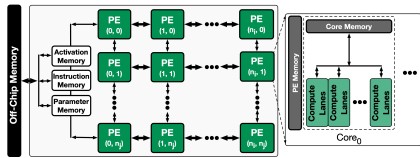

**Figure 2:** An industry-level machine learning accelerator [65].

the instructions (e.g. convolution, pooling, etc.) for execution. The computation stages on such accelerators start by sending a set of activations to the compute lanes, executing them in SIMD manner, and either storing the partial computation results or offloading them back into off-chip memory. Compared to prior works [25, 10, 30], this parameterization is unique—it includes multiple compute lanes per each PE and enables SIMD execution model within each compute lane—and yields a distinct accelerator search space accompanied by an end-to-end simulation framework. More details in Appendix C.

## 3 PROBLEM STATEMENT, TRAINING DATA AND EVALUATION PROTOCOL

Our template-based parameterization maps the accelerator, denoted as $\mathbf{x}$, to a discrete design space, $\mathbf{x} = [\mathbf{x}_1, \mathbf{x}_2, \cdots, \mathbf{x}_K]$, and each $\mathbf{x}_i$ is a discrete-valued variable representing one component of the microarchitectural template, as shown in Table 1 (See Appendix C for the description of other accelerator search spaces studied in our work). A design maybe be infeasible due to various reasons, such as a compilation failure or the limitations of physical implementation, and we denote the set of all such feasibility criterion as Feasible($\mathbf{x}$). The feasibility criterion depends on both the target software and the underlying hardware, and it is not easy to identify if a given $\mathbf{x}$ is infeasible without explicit simulation. We will require our optimization procedure to not only learn the value of the objective function but also to learn to navigate through a sea of infeasible solutions to high-performing feasible solutions $\mathbf{x}^*$ satisfying Feasible($\mathbf{x}^*$) = 1.

Our training dataset $\mathcal{D}$ consists of a modest set of accelerators $\mathbf{x}_i$ that are randomly sampled from the design space and evaluated by the hardware simulator. We partition the dataset $\mathcal{D}$ into two subsets, $\mathcal{D}_{\text{feasible}}$ and $\mathcal{D}_{\text{infeasible}}$. Let $f(\mathbf{x})$ denote the desired objective (e.g., latency, power, etc.) we intend to optimize over the space of accelerators $\mathbf{x}$. We do not possess functional access to $f(\mathbf{x})$, and the optimizer can only access $f(\mathbf{x})$ values for accelerators $\mathbf{x}$ in the feasible partition of the data, $\mathcal{D}_{\text{feasible}}$. For all infeasible accelerators, the simulator does not provide any value of $f(\mathbf{x})$. In addition to satisfying feasibility, the optimizer must handle explicit constraints on parameters such as area and power [13]. In our applications, we impose an explicit area constraint, Area($\mathbf{x}$) $\leq \alpha_0$, though additional explicit constraints are also possible. To account for different constraints, we formulate this task as a constrained optimization problem. Formally:

$$\min_{\mathbf{x}} \ f(\mathbf{x}) \quad \text{s.t.} \quad \text{Area}(\mathbf{x}) \leq \alpha_0, \quad \text{Feasible}(\mathbf{x}) = 1$$
$$\text{on} \ \mathcal{D} = \mathcal{D}_{\text{feasible}} \cup \mathcal{D}_{\text{infeasible}} = \{(\mathbf{x}_1, \mathbf{y}_1), \cdots, (\mathbf{x}_N, \mathbf{y}_N)\} \cup \{\mathbf{x}'_1, \cdots, \mathbf{x}'_{N'}\} \tag{1}$$

While Equation 1 may appear similar to other standard black-box optimization problems, solving it over the space of accelerator designs is challenging due to the large number of infeasible points, the need to handle explicit design constraints, and the difficulty in navigating the non-smooth landscape (See Figure 3 and Figure 11 in the Appendix) of the objective function.

**What makes optimization over accelerators challenging?** Compared to other domains where model-based optimization methods have been applied [5, 57], optimizing accelerators introduces a

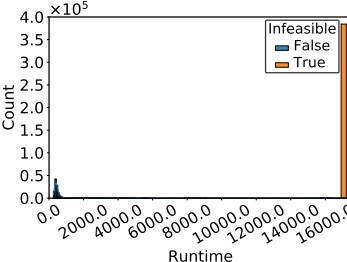 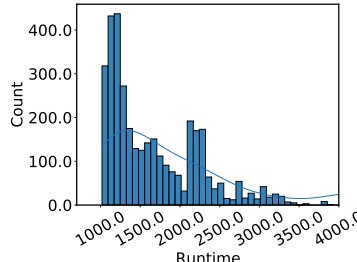

**Figure 3: Left:** histogram of infeasible (right orange bar with large score values) and feasible (left cluster of bars) data points for MobileNetEdgeTPU; **Right:** zoomed-in histogram (different number of bins) focused on feasible points highlighting the variable latencies.

**Table 2:** The description of the applications, their domains, number of (convolutions, depth-wise convolutions, feed-forward) XLA ops, model parameter size, instruction sizes in bytes, number of compute operations.

| Name | Domain | # of XLA Ops (Conv, D/W, FF) | Model Param | Instr. Size | # of Compute Ops. |
|---|---|---|---|---|---|
| **MobileNetEdgeTPU** | Image Class. | (45, 13, 1) | 3.87 MB | 476,736 | 1,989,811,168 |
| **MobileNetV2** | Image Class. | (35, 17, 1) | 3.31 MB | 416,032 | 609,353,376 |
| **MobileNetV3** | Image Class. | (32, 15, 17) | 5.20 MB | 1,331,360 | 449,219,600 |
| **M4** | Object Det. | (32, 13, 2) | 6.23 MB | 317,600 | 3,471,920,128 |
| **M5** | Object Det. | (47, 27, 0) | 2.16 MB | 328,672 | 939,752,960 |
| **M6** | Object Det. | (53, 33, 2) | 0.41 MB | 369,952 | 228,146,848 |
| **U-Net** | Image Seg. | (35, 0, 0) | 3.69 MB | 224,992 | 13,707,214,848 |
| **t-RNN Dec** | Speech Rec. | (0, 0, 19) | 19 MB | 915,008 | 40,116,224 |
| **t-RNN Enc** | Speech Rec. | (0, 0, 18) | 21.62 MB | 909,696 | 45,621,248 |

number of practical challenges. First, accelerator design spaces typically feature a narrow manifold of feasible accelerators within a sea of infeasible points [41, 53, 17], as visualized in Figure 3 and Appendix (Figure 12). While some of these infeasible points can be identified via simple rules (e.g. estimating chip area usage), most infeasible points correspond to failures during compilation or hardware simulation. These infeasible points are generally not straightforward to formulate into the optimization problem and requires simulation [53, 44, 64].

Second, the optimization objective can exhibit high sensitivity to small variations in some architecture parameters (Figure 11b) in some regions of the design space, but remain relatively insensitive in other parts, resulting in a complex optimization landscape. This suggests that optimization algorithms based on local parameter updates (e.g., gradient ascent, evolutionary schemes, etc.) may have a challenging task traversing the nearly flat landscape of the objective, which can lead to poor performance.

**Training dataset.** We used an offline dataset $\mathcal{D}$ of (accelerator parameters, latency) via random sampling from the space of 452M possible accelerator configurations. Our method is only provided with a relatively modest set of feasible points ($\leq 8000$ points) for training, and these points are the *worst-performing* feasible points across the pool of randomly sampled data. This dataset is meant to reflect an easily obtainable and an application-agnostic dataset of accelerators that could have been generated once and stored to disk, or might come from real physical experiments. We emphasize that no assumptions or domain knowledge about the application use case was made during dataset collection. Table 2 depicts the list of target applications, evaluated in this work, includes three variations of MobileNet [23, 50, 27], three in-house industry-level models for object detection (M4, M5, M6; names redacted to prevent anonymity violation), a U-net model [48], and two RNN-based encoder-decoder language models [22, 24, 49, 38]. These applications span the gamut from small models, such as M6, with only 0.4 MB model parameters that demands less on-chip memory, to the medium-sized models ($\geq 5$ MB), such as MobileNetV3 and M4 models, and large models ($\geq 19$ MB), such as t-RNNs, hence requiring larger on-chip memory.

**Evaluation protocol.** To compare state-of-the-art simulator-driven methods and our data-driven method, we limit the number of feasible points (costly to evaluate) that can be used by any algorithm to equal amounts. We still provide infeasible points to any method and leave it up to the optimization method to use it or not. This ensures our comparisons are fair in terms of the amount of data available to each method. However, it is worthwhile to note that in contrast to our method where *worse-quality* data points from small offline dataset are used, the simulator-driven methods have an inherent advantage because they can steer the query process towards the points that are more likely to be better in terms of performance. Following prior work [5, 57, 58], we evaluate each run of a

method by first sampling the top $n = 256$ design candidates according to the algorithm's predictions, evaluating all of these under the ground truth objective function and recording the performance of the best accelerator design. The final reported results is the median of ground truth objective values across five independent runs.

# 4 PRIME: ARCHITECTING ACCELERATORS VIA CONSERVATIVE SURROGATES

As shown in Figure 4, our method first learns a conservative surrogate model of the optimization objective using the offline dataset. Then, it optimizes the learned surrogate using a discrete optimizer. The optimization process does not require access to a simulator, nor to real-world experiments beyond the initial dataset, except when evaluating the final top-performing $n = 256$ designs (Section 3).

**Learning conservative surrogates using logged offline data.** Our goal is to utilize a logged dataset of feasible accelerator designs labeled with the desired performance metric (e.g., latency), $\mathcal{D}_{\text{feasible}}$, and infeasible designs, $\mathcal{D}_{\text{infeasible}}$ to learn a mapping $f_\theta : \mathcal{X} \to \mathbb{R}$, that maps the accelerator configuration $\mathbf{x}$ to its corresponding metric $y$. This learned surrogate can then be optimized by the optimizer. While a straightforward approach for learning such a mapping is to train it via supervised regression, by minimizing the mean-squared error

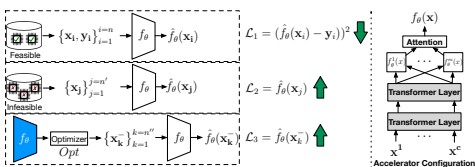

**Figure 4:** Overview of PRIME which trains a conservative surrogate $f_\theta(\mathbf{x}_i)$ using Equation 3. Our neural net model for $f_\theta(\mathbf{x})$ utilizes two transformer layers [59], and a multi-headed architecture which is pooled via a soft-attention layer.

$\mathbb{E}_{\mathbf{x}_i, y_i \sim \mathcal{D}}[(f_\theta(\mathbf{x}_i) - y_i)^2]$, prior work [35, 36, 57] has shown that such predictive models can arbitrarily overestimate the value of an unseen input $\mathbf{x}_i$. This can cause the optimizer to find a solution $\mathbf{x}^*$ that performs poorly in the simulator but looks promising under the learned model. We empirically validate this overestimation hypothesis and find it to confound the optimizer in on our problem domain as well (See Figure 13 in Appendix). To prevent overestimated values at unseen inputs from confounding the optimizer, we build on COMs [57] and train $f_\theta(\mathbf{x})$ with an additional term that explicitly maximizes the function value $f_\theta(\mathbf{x})$ at unseen $\mathbf{x}$ values. Such unseen designs $\mathbf{x}$, where the learned function $f_\theta(\mathbf{x})$ is likely to be overestimated, are "negative mined" by running a few iterations of an approximate stochastic optimization procedure that aims to maximize $f_\theta$ in the inner loop. This procedure is analogous to adversarial training [19]. Equation 2 formalizes this objective:

$$\theta^* := \arg\min_\theta \ \mathcal{L}(\theta) := \mathbb{E}_{\mathbf{x}_i, y_i \sim \mathcal{D}_{\text{feasible}}} \left[ (f_\theta(\mathbf{x}_i) - y_i)^2 \right] - \alpha \mathbb{E}_{\mathbf{x}_i^- \sim \text{Opt}(f_\theta)} \left[ f_\theta(\mathbf{x}_i^-) \right]. \quad (2)$$

$\mathbf{x}_i^-$ denotes the negative samples produced from an optimizer $\text{Opt}(\cdot)$ that attempts to maximize the current learned model, $f_\theta$. We will discuss our choice of $\text{Opt}$ in the Appendix Section B.

**Incorporating design constraints via infeasible points.** While prior work [57] simply optimizes Equation 2 to learn a surrogate, this is not enough when optimizing over accelerators, as we will also show empirically (Appendix A.1). This is because explicit negative mining does not provide any information about accelerator design constraints. Fortunately, this information is provided by infeasible points, $\mathcal{D}_{\text{infeasible}}$. The training procedure in Equation 2 provides a simple way to do incorporate such infeasible points: we simply incorporate $\mathbf{x}_i' \sim \mathcal{D}_{\text{infeasible}}$ as additional negative samples and maximize the prediction at these points. This gives rise to our final objective:

$$\min_\theta \ \mathcal{L}^{\text{inf}}(\theta) := \mathcal{L}(\theta) - \beta \mathbb{E}_{\mathbf{x}_i' \sim \mathcal{D}_{\text{infeasible}}} \left[ f_\theta(\mathbf{x}_i') \right] \quad (3)$$

**Multi-model optimization and zero-shot generalization.** One of the central benefits of a data-driven approach is that it enables learning powerful surrogates that generalize over the space of applications, potentially being effective for new unseen application domains. In our experiments, we evaluate PRIME on designing accelerators for multiple applications denoted as $k = 1, \cdots, K$, jointly or for a novel unseen application. In this case, we utilized a dataset $\mathcal{D} = \{\mathcal{D}_1, \cdots, \mathcal{D}_K\}$, where each $\mathcal{D}_k$ consists of a set of accelerator designs, annotated with the latency value and the feasibility criterion for a given application $k$. While there are a few overlapping designs in different parts of the dataset annotated for different applications, most of the designs only appear in one part. To train a single conservative surrogate $f_\theta(\mathbf{x})$ for multiple applications, we extend the training procedure in Equation 3 to incorporate *context vectors* $\mathbf{c}_k \in \mathbb{R}^d$ for various applications driven by a list of application properties in Table 2. The learned function in this setting is now conditioned on the context $f_\theta(\mathbf{x}, \mathbf{c}_k)$. We train $f_\theta$ via the objective in Equation 3, but in expectation over all the contexts

and their corresponding datasets: $\min_\theta \mathbb{E}_{k \sim [K]} \left[ \mathcal{L}_k^{\inf}(\theta) \right]$. Once such a contextual surrogate is learned, we can either optimize the average surrogate across a set of contexts $\{\mathbf{c}_i, \mathbf{c}_2, \cdots, \mathbf{c}_n\}$ to obtain an accelerator that is optimal for multiple applications simultaneously on an average ("multi-model" optimization), or optimize this contextual surrogate for a novel context vector, corresponding to an unseen application ("zero-shot" generalization). In this case, PRIME is not allowed to train on any data corresponding to this new unseen application. While such zero-shot generalization might appear surprising at first, note that the context vectors are not simply one-hot vectors, but consist of parameters with semantic information, which the surrogate can generalize over.

**Learned conservative surrogate optimization.** Prior work [64] has shown that the most effective optimizers for accelerator design are meta-heuristic/evolutionary optimizers. We therefore choose to utilize, firefly [62, 63, 39] to optimize our conservative surrogate. This algorithm maintains a set of optimization candidates (a.k.a. "fireflies") and jointly update them towards regions of low objective value, while adjusting their relative distances appropriately to ensure multiple high-performing, but diverse solutions. We discuss additional details in Appendix B.1.

**Cross validation: which model and checkpoint should we evaluate?** Similarly to supervised learning, models trained via Equation 3 can overfit, leading to poor solutions. Thus, we require a procedure to select which hyperparameters and checkpoints should actually be used for the design. This is crucial, because we cannot arbitrarily evaluate as many models as we want against the simulator. While effective methods for model selection have been hard to develop in offline optimization [57, 58], we devised a simple scheme using a validation set for choosing the values of $\alpha$ and $\beta$ (Equation 3), as well as which checkpoint to utilize for generating the design. For each training run, we hold out the best 20% of the points out of the training set and use them *only* for cross-validation as follows. Typical cross-validation strategies in supervised learning involve tracking validation error (or risk), but since our model is trained conservatively, its predictions may not match the ground truth, making such validation risk values unsuitable for our use case. Instead, we track Kendall's ranking correlation between the predictions of the learned model $f_\theta(\mathbf{x}_i)$ and the ground truth values $y_i$ (Appendix B) for the held-out points for each run. We pick values of $\alpha$, $\beta$ and the checkpoint that attain the highest validation ranking correlation. We present the pseudo-code for PRIME (Algorithm 1) and implementation details in Appendix B.1.

## 5 RELATED WORK

Optimizing hardware accelerators has become more important recently. Prior works [45, 28, 56, 41, 8, 34, 4, 3, 25, 60, 61] mainly rely on expensive-to-query hardware simulators to navigate the search space and/or target single-application accelerators. For example, HyperMapper [41] targets compiler optimization for FPGAs by continuously interacting with the simulator in a design space with relatively few infeasible points. Mind Mappings [25], optimizes software mappings to a fixed hardware provided access to millions of feasible points and throws away infeasible points during learning. MAGNet [60] uses a combination of pruning heuristics and online Bayesian optimization to generate accelerators for image classification models in a single-application setting. AutoDNNChip [61] uses two-level online optimization to generate customized accelerators for ASIC and FPAG platforms. In contrast, PRIME , does not only learn a surrogate using offline data but can also leverage information from infeasible points and can work with just a few thousand feasible points. In addition, we devise a contextual version of PRIME that is effective in designing accelerators that are jointly optimized for multiple applications, different from prior work. Finally, to our knowledge, our work, is the first to demonstrate generalization to unseen applications for accelerator design, outperforming state-of-the-art online methods.

A popular approach for solving black-box optimization problems is model-based optimization (MBO) [55, 51, 54]. Most of these methods fail to scale to high-dimensions, and have been extended with neural networks [55, 54, 31, 16, 15, 2, 1, 40]. While these methods work well in the active setting, they are susceptible to out-of-distribution inputs [58] in the offline, data-driven setting. To prevent this, offline MBO methods that constrain the optimizer to the manifold of valid, in-distribution inputs have been developed Brookes et al. [5], Fannjiang & Listgarten [12], Kumar & Levine [35]. However, modeling the manifold of valid inputs can be challenging for accelerators. PRIME dispenses with the need for generative modeling, while still avoiding out-of-distribution inputs. PRIME builds on "conservative" offline RL and offline MBO methods that train robust surrogates [36, 57]. However, unlike these approaches, PRIME can handle constraints by learning from infeasible data and utilizes a better optimizer (See Appendix Table 7 for a comparison). In addition, while prior works area mostly

**Table 3:** Optimized objective values (i.e., latency in milliseconds) obtained by various methods for the task of learning accelerators specialized to a given application. Lower latency is better. **From left to right**: our method, online Bayesian optimization ("Bayes Opt"), online evolutionary algorithm ("Evolutionary"), and the best design in the training dataset. On average (last row), PRIME improves over the best in the dataset by 2.46× (up to 6.69× in t-RNN Dec) and outperforms best online optimization methods by 1.54× (up to 6.62× in t-RNN Enc). The best accelerator configurations identified is highlighted in bold.

| Application | PRIME | Online Optimization | | | $\mathcal{D}$ (Best in Training) |
| | | Bayes Opt | Evolutionary | MBO | |
|---|---|---|---|---|---|
| MobileNetEdgeTPU | **298.50** | 319.00 | 320.28 | 332.97 | 354.13 |
| MobileNetV2 | **207.43** | 240.56 | 238.58 | 244.98 | 410.83 |
| MobileNetV3 | **454.30** | 534.15 | 501.27 | 535.34 | 938.41 |
| M4 | **370.45** | 396.36 | 383.58 | 405.60 | 779.98 |
| M5 | 208.21 | 201.59 | **198.86** | 219.53 | 449.38 |
| M6 | 131.46 | 121.83 | 120.49 | **119.56** | 369.85 |
| U-Net | **740.27** | 872.23 | 791.64 | 888.16 | 1333.18 |
| t-RNN Dec | **132.88** | 771.11 | 770.93 | 771.70 | 890.22 |
| t-RNN Enc | **130.67** | 865.07 | 865.07 | 866.28 | 584.70 |
| **Geomean of PRIME's Improvement** | 1.0× | **1.58×** | **1.54×** | **1.61×** | **2.46×** |

restricted to a single application, we show that PRIME is effective in multi-task optimization and zero-shot generalization.

## 6 EXPERIMENTAL EVALUATION

Our evaluations aim to answer the following questions: **Q(1)** Can PRIME design accelerators tailored for a given application that are better than the best observed configuration in the training dataset, and comparable to or better than state-of-the-art simulation-driven methods under a given simulator-query budget? **Q(2)** Does PRIME reduce the total simulation time compared to other methods? **Q(3)** Can PRIME produce hardware accelerators for a family of different applications? **Q(4)** Can PRIME trained for a family of applications extrapolate to designing a high-performing accelerator for a new, unseen application, thereby enabling data reuse? Additionally, we ablate various properties of PRIME (Appendix A.6) and evaluate its efficacy in designing accelerators with distinct dataflow architectures, with a larger search space (up to $2.5 \times 10^{114}$ possible candidates).

**Baselines and comparisons.** We compare PRIME against three online optimization methods that actively query the simulator: **(1)** evolutionary search with the firefly optimizer [64] ("Evolutionary"), which is the shown to outperform other online methods for accelerator design; **(2)** Bayesian Optimization ("Bayes Opt") [18], **(3)** MBO [1]. In all the experiments, we grant all the methods the same number of feasible points. Note that our method do not get to select these points, and use the same exact offline points across all the runs, while the online methods can actively select which points to query, and therefore require new queries for every run. "$\mathcal{D}$(Best in Training)" denotes the best latency value in the training dataset used in PRIME. We also present ablation results with different components of our method removed in Appendix A.6, where we observe that utilizing both infeasible points and negative sampling are generally important for attaining good results. Appendix A.1 presents additional comparisons to COMs Trabucco et al. [57]—which

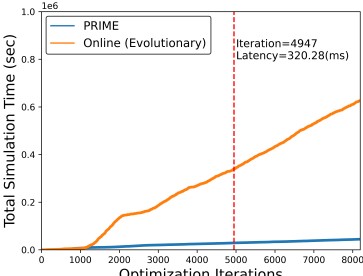

**Figure 5:** Comparing the total simulation time of PRIME (for PRIME this is the total time for a forward-pass through the trained surrogate on a CPU) and evolutionary method on MobileNet-EdgeTPU. PRIME only requires about **7%** of the total simulation time of the online method.

only obtains negative samples via gradient ascent on the learned surrogate and does not utilize infeasible points—and P3BO Angermueller et al. [2]—an state-of-the-art online method in biology.

**Architecting application-specific accelerators.** We first evaluate PRIME in designing specialized accelerators for each of the applications in Table 2. We train a conservative surrogate using the method in Section 4 on the logged dataset for each application separately. The area constraint $\alpha$ (Equation 1) is set to $\alpha = 29$ mm$^2$, a realistic budget for accelerators [64]. Table 3 summarizes the results. On average, the best accelerators designed by PRIME outperforms the best accelerator configuration in the training dataset (last row Table 3), by 2.46×. PRIME also outperforms the accelerators in the best online method by 1.54× (up to 5.80× and 6.62× in t-RNN Dec and t-RNN

**Table 4:** Optimized average latency (the lower, the better) across multiple applications (up to ten applications) from diverse domains by PRIME and best online algorithms (Evolutionary and MBO) under different area constraints. Each row show the (Best, Median) of average latency across five runs. The geometric mean of PRIME's improvement over other methods (last row) indicates that PRIME is at least 21% better.

| Applications | Area | PRIME (Ours) | Evolutionary (Online) | MBO (Online) |
|---|---|---|---|---|
| MobileNet (EdgeTPU, V2, V3) | 29 mm$^2$ | (310.21, 334.70) | (**315.72**, **325.69**) | (342.02, 351.92) |
| MobileNet (V2, V3), M5, M6 | 29 mm$^2$ | (**268.47**, **271.25**) | (288.67, 288.68) | (295.21, 307.09) |
| MobileNet (EdgeTPU, V2, V3), M4, M5, M6 | 29 mm$^2$ | (**311.39**, **313.76**) | (314.31, 316.65) | (321.48, 339.27) |
| MobileNet (EdgeTPU, V2, V3), M4, M5, M6, U-Net, t-RNN-Enc | 29 mm$^2$ | (**305.47**, **310.09**) | (404.06, 404.59) | (404.06, 412.90) |
| MobileNet (EdgeTPU, V2, V3), M4, M5, M6, t-RNN-Enc | 100 mm$^2$ | (**286.45**, **287.98**) | (404.25, 404.59) | (404.06, 404.94) |
| MobileNet (EdgeTPU, V2, V3), M4, M5, M6, t-RNN (Dec, Enc) | 29 mm$^2$ | (**426.65**, **426.65**) | (586.55, 586.55) | (626.62, 692.61) |
| MobileNet (EdgeTPU, V2, V3), M4, M5, M6, U-Net, t-RNN (Dec, Enc) | 100 mm$^2$ | (**383.57**, **385.56**) | (518.58, 519.37) | (526.37, 530.99) |
| **Geomean of PRIME's Improvement** | — | (**1.0×**, **1.0×**) | (**1.21×**, **1.20×**) | (**1.24×**, **1.27×**) |

Enc, respectively). Moreover, perhaps surprisingly, PRIME generates accelerators that are better than all the online optimization methods in 7/9 domains, and performs on par in several other scenarios (on average only 6.8% slowdown compared to the best accelerator with online methods in M5 and M6). These results indicates that offline optimization of accelerators using PRIME can be more data-efficient compared to online methods with active simulation queries. To answer **Q(2)**, we compare the total simulation time of PRIME and the best evolutionary approach from Table 3 on the MobileNetEdgeTPU domain. On average, not only that PRIME outperforms the best online method that we evaluate, but also considerably reduces the total simulation time by 93%, as shown in Figure 5. Even the total simulation time to the first occurrence of the final design that is eventually returned by the online methods is about 11× what PRIME requires to fine a better design. This indicates that data-driven PRIME is much more preferred in terms of the performance-time trade-off. The fact that our offline approach PRIME outperforms the online evolutionary method (and also other state-of-the-art online MBO methods; see Table 8) is surprising, and we suspect this is because online methods get stuck early-on during optimization, while utilizing offline data allows us PRIME to find better solutions via generalization (see Appendix B.1.1).

**Architecting accelerators for multiple applications.** To answer **Q(3)**, we evaluate the efficacy of the contextual version of PRIME in designing an accelerator that attains the lowest latency averaged over a set of application domains. As discussed previously, the training data used does not label a given accelerator with latency values corresponding to each application, and thus, PRIME must extrapolate accurately to estimate the latency of an accelerator for a context it is not paired with in the training dataset. This also means that PRIME cannot simply return the accelerator with the best average latency and must run non-trivial optimization. We evaluate our method in seven different scenarios (Table 4), comprising various combinations of models from Table 2 and under different area constraints, where the smallest set consists of the three MobileNet variants and the largest set consists of nine models from image classification, object detection, image segmentation, and speech

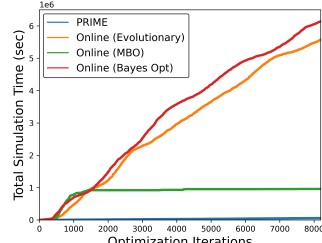

**Figure 6:** Comparing the total simulation time needed by PRIME and online methods on seven models (Area $\leq 100\text{mm}^2$). PRIME only requires about 1%, 6%, and 0.9% of the total simulation time of Evolutionary, MBO, and Bayes Opt, respectively, although PRIME outperforms the best online method by 41%.

recognition. This scenario is also especially challenging for online methods since the number of jointly feasible designs is expected to drop significantly as more applications are added. For instance, for the case of the MobileNet variants, the training dataset only consists of a few (20-30) accelerator configurations that are jointly feasible and high-performing (Appendix B.2—Figure 9).

Table 4 shows that, on average, PRIME finds accelerators that outperform the best online method by 1.2× (up to 41%). While PRIME performs similar to online methods in the smallest three-model scenario (first row), it outperforms online methods as the number of applications increases and the set of applications become more diverse. In addition, comparing with the best jointly feasible design point across the target applications, PRIME finds significantly better accelerators (3.95×). Finally, as the number of model increases the total simulation time difference between online methods and PRIME further widens (Figure 6). These results indicate that PRIME is effective in designing accelerators jointly optimized across multiple applications while reusing the same dataset as for the single-task, and scales more favorably than its simulation-driven counterparts. Appendix A.4 expounds the details of the designed accelerators for nine applications, comparing our method and the best online method.

**Table 5:** Optimized objective values (i.e., latency in milliseconds) under zero-shot setting. Lower latency is better. From left to right: the applications used to train the surrogate model in PRIME the target applications for which the accelerator is optimized for, the area constraint of the accelerator, PRIME's (best, median) latency, and best online method's (best, median) latency. PRIME does not use any additional data from the target applications. On average (last row), PRIME yields optimized accelerator for target applications (with zero query to the target applications' dataset) with $1.26\times$ (up to $1.66\times$) lower latency over the best online method. The best accelerator configurations identified is highlighted in bold.

| Train Applications | Test Applications | Area | PRIME (Ours) | Evolutionary (Online) |
|---|---|---|---|---|
| MobileNet (EdgeTPU, V3) | MobileNetV2 | 29 mm$^2$ | (**311.39**, **313.76**) | (314.31, 316.65) |
| MobileNet (V2, V3), M5, M6 | MobileNetEdge, M4 | 29 mm$^2$ | (357.05, 364.92) | (**354.59**, **357.29**) |
| MobileNet (EdgeTPU, V2, V3), M4, M5, M6, t-RNN Enc | U-Net, t-RNN Dec | 29 mm$^2$ | (**745.87**, **745.91**) | (1075.91, 1127.64) |
| MobileNet (EdgeTPU, V2, V3),M4, M5, M6, t-RNN Enc | U-Net, t-RNN Dec | 100 mm$^2$ | (**517.76**, **517.89**) | (859.76, 861.69) |
| **Geomean of PRIME's Improvement** | — | — | (**1.0×**, **1.0×**) | (**1.24×**, **1.26×**) |

**Table 6:** Optimized objective values (i.e. total number of cycles) for two different dataflow architectures, NVDLA-style [42] and ShiDianNao-style [10], across three classes of applications. The maximum search space for the studied accelerators are $\approx 2.5\times10^{114}$. PRIME generalizes to other classes of accelerators with larger search space and outperforms the best online method by $1.06\times$ and the best data seen in training by $3.75\times$ (last column). The best accelerator configurations is highlighted in bold.

| Applications | Dataflow | PRIME | Evolutionary (Online) | $\mathcal{D}$ (Best in Training) |
|---|---|---|---|---|
| MobileNetV2 | NVDLA | $\mathbf{2.51\times10^7}$ | $2.70\times10^7$ | $1.32\times10^8$ |
| MobileNetV2 | ShiDianNao | $\mathbf{2.65\times10^7}$ | $2.84\times10^7$ | $1.27\times10^8$ |
| ResNet50 | NVDLA | $\mathbf{2.83\times10^8}$ | $3.13\times10^8$ | $1.63\times10^9$ |
| ResNet50 | ShiDianNao | $\mathbf{3.44\times10^8}$ | $3.74\times10^8$ | $2.05\times10^9$ |
| Transformer | NVDLA | $\mathbf{7.8\times10^8}$ | $7.8\times10^8$ | $1.3\times10^9$ |
| Transformer | ShiDianNao | $\mathbf{7.8\times10^8}$ | $7.8\times10^8$ | $1.5\times10^9$ |
| **Geomean of PRIME's Improvement** | — | **1.0×** | **1.06×** | **3.75×** |

**Accelerating previously unseen applications ("zero-shot" optimization).** Finally, we answer **Q(4)** by demonstrating that our data-driven offline method, PRIME enables effective data reuse by using logged accelerator data from a set of applications to design an accelerator for an unseen new application, without requiring any training on data from the new unseen application(s). We train a contextual version of PRIME using a set of "training applications" and then optimize an accelerator using the learned surrogate with different contexts corresponding to "test applications," without any additional query to the test application dataset. Table 5 shows, on average, PRIME outperforms the best online method by $1.26\times$ (up to 66%) and only 2% slowdown in 1/4 cases. Note that the difference in performance increases as the number of training applications increases. These results show the effectiveness of PRIME in the zero-shot setting (more results in Appendix A.5).

**Applying PRIME on other accelerator architectures and dataflows.** Finally, to assess the the generalizability of PRIME to other accelerator architectures Kao et al. [30], we evaluate PRIME to optimize latency of two style of dataflow accelerators—NVDLA-style and ShiDianNao-style—across three applications (Appendix C details the methodology). As shown in Table 6, PRIME outperforms the online evolutionary method by 6% and improves over the best point in the training dataset by $3.75\times$. This demonstrates the efficacy of PRIME with different dataflows and large design spaces.

## 7 DISCUSSION

In this work, we present a data-driven offline optimization method, PRIME to automatically architect hardware accelerators. Our method learns a conservative surrogate of the objective function by leveraging infeasible data points to better model the desired objective function of the accelerator using a one-time collected dataset of accelerators, thereby alleviating the need for time-consuming simulation. Our results show that, on average, our method outperforms the best designs observed in the logged data by $2.46\times$ and improves over the best simulator-driven approach by about $1.54\times$. In the more challenging setting of designing accelerators jointly optimal for multiple applications or for new, unseen applications, zero-shot, PRIME outperforms simulator-driven methods by $1.2\times$, while reducing the total simulation time by 99%. The efficacy of PRIME highlights the potential for utilizing the logged offline data in an accelerator design pipeline. While PRIME outperforms the online methods we utilize, in principle, a strong online method can be devised by running PRIME in the inner loop. Our goal is to not advocate that offline methods must replace online methods, but that training a strong offline optimization algorithm on offline datasets of low-performing designs can be a highly effective ingredient in hardware accelerator design.

## ACKNOWLEDGEMENTS

We thank the "Learn to Design Accelerators" team at Google Research and the Google EdgeTPU team for their invaluable feedback and suggestions. In addition, we extend our gratitude to the Vizier team, Christof Angermueller, Sheng-Chun Kao, Samira Khan, Stella Aslibekyan, and Xinyang Geng for their help with experiment setups and insightful comments.

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

# Appendices

## A  ADDITIONAL EXPERIMENTS

In this section, we present additional experiments compared to the method of Trabucco et al. [57]), present some additional results obtained by jointly optimizing multiple applications (Appendix A.2), provide an analysis of the designed accelerators (Appendix A.4) and finally, discuss how our trained conservative surrogate can be used with a different evaluation time constraint (Appendix A.3).

### A.1  COMPARISON TO OTHER BASELINE METHODS

**Comparison to COMs.** In this section, we perform a comparative evaluation of PRIME to the COMs method Trabucco et al. [57]. Like several offline reinforcement learning algorithms [36], our method, PRIME and COMs are based on the key idea of learning a conservative surrogate of the desired objective function, such that it does not overestimate the value of unseen data points, which prevents the optimizer from finding accelerators that appear promising under the learned model but are not actually promising under the actual objective. The key differences between our method and COMs are: **(1)** PRIME uses an evolutionary optimizer $(\mathrm{Opt}(\cdot))$ for negative sampling compared to gradient ascent of COMs, which can be vastly beneficial in discrete design spaces as our results show empirically, **(2)** PRIME can explicitly learn from infeasible data points provided to the algorithm, while COMs does not have a mechanism to incorporate the infeasible points into the learning of surrogate. To further assess the importance of these differences in practice, we run COMs on three tasks from Table 3, and present a comparison our method, COMs, and Standard method in Table 7. The "Standard" method represents a surrogate model without utilizing any infeasible points. On average, PRIME outperforms COMs by $1.17\times$ (up to $1.24\times$ in M6).

**Table 7:** Optimized objective values (i.e., latency in milliseconds) obtained by PRIME and COMs [57] when optimizing over single applications (MobilenetV2, MobilenetV3 and M6), extending Table 3. Note that PRIME outperforms COMs. However, COMs improves over baseline "Standard" method (last column).

| Application | PRIME (Ours) | COMs | Standard |
|---|---|---|---|
| MobileNetV2 | **207.43** | 251.58 | 374.52 |
| MobileNetV3 | **454.30** | 485.66 | 575.75 |
| M6 | **131.46** | 163.94 | 180.24 |
| **Geomean of PRIME's Improvement** | **1.0×** | **1.17×** | **1.46×** |

**Comparison to generative offline MBO methods.** We provide a comparison between PRIME and prior offline MBO methods based on generative models [35]. We evaluate model inversion networks (MINs) [35] on our accelerator data. However, we were unable to train a discrete objective-conditioned GAN model to 0.5 discriminator accuracy on our offline dataset, and often observed a collapse of the discriminator. As a result, we trained a $\delta-$VAE [46], conditioned on the objective function (i.e., latency). A standard VAE [33] suffered from posterior collapse and thus informed our choice of utilizing a $\delta-$VAE. The latent space of a trained objective-conditioned $\delta-$VAE corresponding to accelerators on a held-out validation dataset (not used for training) is visualized in the t-SNE plot in the figure on the right. This is a 2D t-SNE of the accelerators configurations (§Table 1). The color of a point denotes the latency value of the corresponding accelerator configuration, partitioned into three bins. Observe that while we would expect these objective conditioned models to disentangle accelerators with different objective values in the latent space, the

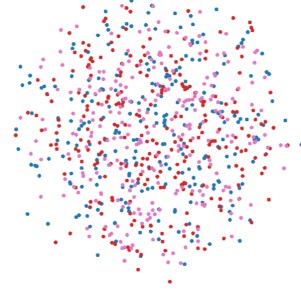

models we trained did not exhibit such a structure, which will hamper optimization. While our method PRIME could also benefit from a generative optimizer (i.e., by using a generative optimizer in place of $\mathrm{Opt}(\cdot)$ with a conservative surrogate), we leave it for future work to design effective generative optimizers on the accelerator manifold.

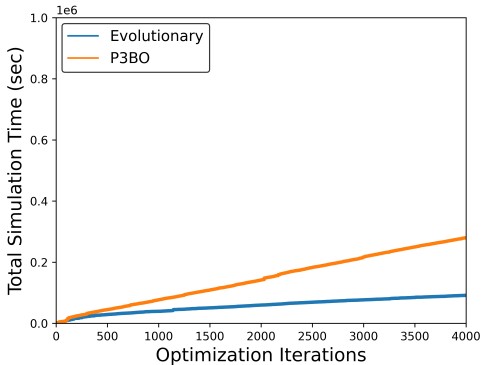

**Figure 7:** Comparing the total simulation time needed by the P3BO and Evolutionary method on MobileNet-EdgeTPU. Note that, not only the total simulation time of P3BO is around $3.1\times$ higher than the Evolutionary method, but also the latency of final optimized accelerator is around 18% for MobileNetEdgeTPU. The total simulation time of our method is around 7% of the Evolutionary method (See Figure 5).

**Comparison to P3BO.** We perform a comparison against P3BO method, the state-of-the-arts online method in biology [2]. On average, PRIME outperforms the P3BO method by $2.5\times$ (up to $8.7\times$ in U-Net). In addition, we present the comparison between the total simulation runtime of the P3BO and Evolutionary methods in Figure 7. Note that, not only the total simulation time of P3BO is around $3.1\times$ higher than the Evolutionary method, but also the latency of final optimized accelerator is around 18% for MobileNetEdgeTPU. On the other hand, the total simulation time of PRIME for the task of accelerator design for MobileNetEdgeTPU is lower than both methods (only 7% of the Evolutionary method as shown in Figure 5).

**Table 8:** Optimized objective values (i.e., latency in milliseconds) obtained by PRIME and P3BO [2] when optimizing over single applications (MobileNetEdgeTPU, M4, t-RNN Dec, t-RNN Enc, and U-Net). On average, PRIME outperforms P3BO by $2.5\times$.

| Application | PRIME (Ours) | P3BO |
|---|---|---|
| MobileNetEdgeTPU | **298.50** | 376.02 |
| M4 | **370.45** | 483.39 |
| U-Net | **740.27** | 771.70 |
| t-RNN Dec | **132.88** | 865.12 |
| t-RNN Enc | **130.67** | 1139.48 |
| **Geomean of PRIME's Improvement** | **1.0×** | **2.5×** |

## A.2 Learned Surrogate Model Reuse for Accelerator Design

Extending our results in Table 4, we present another variant of optimizing accelerators jointly for multiple applications. In that scenario, the learned surrogate model is reused to architect an accelerator for a subset of applications used for training. We train a contextual conservative surrogate on the variants of MobileNet (Table 2) as discussed in Section 4, but generated optimized designs by only optimizing the average surrogate on only two variants of MobileNet (MobileNetEdgeTPU and MobileNetV2). This tests the ability of our approach PRIME to provide a general contextual conservative surrogates, that can be trained *only once* and optimized multiple times with respect to different subsets of applications. Observe in Table 9, PRIME architects high-performing accelerator configurations (better than the best point in the dataset by $3.29\times$ – last column) while outperforming the online optimization methods by 7%.

## A.3 Learned Surrogate Model Reuse under Different Design Constraint

We also test the robustness of our approach in handling variable constraints at test-time such as different chip area budget. We evaluate the learned conservative surrogate trained via PRIME under a reduced value of the area threshold, $\alpha$, in Equation 1. To do so, we utilize a variant of rejection sampling – we take the learned model trained for a default area constraint $\alpha = 29 \, \text{mm}^2$ and then reject

**Table 9:** Optimized objective values (i.e., latency in milliseconds) obtained by our PRIME when using the jointly optimized model on three variants of MobileNets and use for MobileNetEdgeTPU and MobileNetV2 for different dataset configurations. PRIME outperforms the best online method by **7%** and finds an accelerator that is **3.29×** better than the best accelerator in the training dataset (last row). The best accelerator configuration is highlighted in bold.

| Applications | PRIME | | | Standard | Online Optimization | | $\mathcal{D}$ (Best in Training) |
| | All | -Opt | -Infeasible | | Bayes Opt | Evolutionary | |
|---|---|---|---|---|---|---|---|
| (MobileNetEdgeTPU, MobileNetV2) | **253.85** | 297.36 | 264.85 | 341.12 | 275.21 | 271.71 | 834.68 |

**Table 10:** Optimized objective values (i.e., latency in milliseconds) obtained by various methods for the task of learning accelerators specialized to MobileNetEdgeTPU under chip area budget constraint 18 mm$^2$ reusing the already learned model by our method for MobileNetEdgeTPU (shown in Table 3). Lower latency/runtime is better. From left to right: our method, our method without negative sampling ("PRIME-Opt") and without utilizing infeasible points ("PRIME-Infeasible"), standard surrogate ("Standard"), online Bayesian optimization ("Bayes Opt"), online evolutionary algorithm ("Evolutionary") and the best design in the training dataset. Note that PRIME improves over the best in the dataset by 12%, outperforms the best online optimization method by 4.4%. The best accelerator configuration is highlighted in bold.

| Applications | PRIME | | | Standard | Online Optimization | | $\mathcal{D}$ (Best in Training) |
| | All | -Opt | -Infeasible | | Bayes Opt | Evolutionary | |
|---|---|---|---|---|---|---|---|
| MobileNetEdgeTPU, Area $\leq$ 18 mm$^2$ | **315.15** | 433.81 | 351.22 | 470.09 | 331.05 | 329.13 | 354.13 |

**Table 11:** Per application latency for the best accelerator design suggested by PRIME and the Evolutionary method according to Table 4 for multi-task accelerator design (nine applications and area constraint 100 mm$^2$). PRIME outperforms the Evolutionary method by 1.35×.

| Applications | Latency (ms) | | Improvement of PRIME over Evolutionary |
| | PRIME | Evolutionary (Online) | |
|---|---|---|---|
| MobileNetEdgeTPU | **288.38** | 319.98 | 1.10× |
| MobileNetV2 | **216.27** | 255.95 | 1.18× |
| MobileNetV3 | **487.46** | 573.57 | 1.17× |
| M4 | **400.88** | 406.28 | 1.01× |
| M5 | 248.18 | **239.18** | 0.96× |
| M6 | 164.98 | **148.83** | 0.90× |
| U-Net | 1268.73 | **908.86** | 0.71× |
| t-RNN Dec | **191.83** | 862.14 | 5.13× |
| t-RNN Enc | **185.41** | 952.44 | 4.49× |
| **Average (Latency in ms)** | **383.57** | **518.58** | **1.35×** |

all optimized accelerator configurations which do not satisfy a reduces area constraint: $\mathrm{Area}(\mathbf{x}) \leq \alpha_0 = 18 \text{ mm}^2$. Table 10 summarizes the results for this scenario for the MobileNetEdgeTPU [23] application under the new area constraint ($\alpha = 18 \text{ mm}^2$). A method that produces diverse designs which are both high-performing and are spread across diverse values of the area constraint are expected to perform better. As shown in Table 10, PRIME provides better accelerator than the best online optimization from scratch with the new constraint value by 4.4%, even when PRIME does not train its conservative surrogate with this unseen test-time design constraint. Note that, when the design constraint changes, online methods generally need to restart the optimization process from scratch and undergo costly queries to the simulator. This would impose additional overhead in terms of total simulation time (§ Figure 5 and Figure 6). However, the results in Table 10 shows that our learned surrogate model can be reused under different test-time design constraint eliminating additional queries to the simulator.

## A.4 ANALYSIS OF DESIGNED ACCELERATORS

In this section, we overview the best accelerator configurations that PRIME and the Evolutionary method identified for multi-task accelerator design (See Table 4), when the number of target applications are nine and the area constraint is set to 100 mm$^2$. The average latencies of the best accelerators found by PRIME and the Evolutionary method across nine target applications are **383.57 ms** and **518.58 ms**, respectively. In this setting, our method outperforms the best online method by **1.35×**.

**Table 12:** Optimized accelerator configurations (See Table 1) found by PRIME and the Evolutionary method for multi-task accelerator design (nine applications and area constraint 100 mm$^2$). Last row shows the accelerator area in mm$^2$. PRIME reduces the overall chip area usage by 1.97×. The difference in the accelerator configurations are shaded in gray.

| Accelerator Parameter | Parameter Value | |
| --- | --- | --- |
| | PRIME | Evolutionary (Online) |
| # of PEs-X | 4 | 4 |
| # of PEs-Y | 6 | 8 |
| # of Cores | 64 | 128 |
| # of Compute Lanes | 4 | 6 |
| PE Memory | 2,097,152 | 1,048,576 |
| Core Memory | 131,072 | 131,072 |
| Instruction Memory | 32,768 | 8,192 |
| Parameter Memory | 4,096 | 4,096 |
| Activation Memory | 512 | 2,048 |
| DRAM Bandwidth (Gbps) | 30 | 30 |
| **Chip Area (mm$^2$)** | **46.78** | **92.05** |

Table 11 shows per application latencies for the accelerator suggested by our method and the Evolutionary method. The last column shows the latency improvement of PRIME over the Evolutionary method. Interestingly, while the latency of the accelerator found by our method for MobileNetEdgeTPU, MobileNetV2, MobileNetV3, M4, t-RNN Dec, and t-RNN Enc are better, the accelerator identified by the online method yields lower latency in M5, M6, and U-Net.

To better understand the trade-off in design of each accelerator designed by our method and the Evolutionary method, we present all the accelerator parameters (See Table 1) in Table 12. The accelerator parameters that are different between each of the designed accelerator are shaded in gray (e.g. # of PEs-Y, # of Cores, # of Compute Lanes, PE Memory, Instruction Memory, and Activation Memory). Last row of Table 12 depicts the overall chip area usage in mm$^2$. PRIME not only outperforms the Evolutionary algorithm in reducing the average latency across the set of target applications, but also reduces the overall chip area usage by **1.97×**. Studying the identified accelerator configuration, we observe that PRIME trade-offs compute ( **64 cores vs. 128 cores** ) for larger PE memory size ( **2,097,152 vs. 1,048,576** ). These results show that PRIME favors PE memory size to accommodate for the larger memory requirements in t-RNN Dec and t-RNN Enc (See Table 2 Model Parameters) where large gains lie. Favoring larger on-chip memory comes at the expense of lower compute power in the accelerator. This reduction in the accelerator's compute power leads to higher latency for the models with large number of compute operations, namely M5, M6, and U-Net (See last row in Table 2). M4 is an interesting case where both compute power and on-chip memory is favored by the model (6.23 MB model parameters and 3,471,920,128 number of compute operations). This is the reason that the latency of this model on both accelerators, designed by our method and the Evolutionary method, are comparable (400.88 ms in PRIME vs. 406.28 ms in the online method).

## A.5 COMPARISON WITH ONLINE METHODS IN ZERO-SHOT SETTING

We evaluated the Evolutionary (online) method under two protocols for the last two rows of Table 5: first, we picked the best designs (top-performing 256 designs similar to the PRIME setting in Section 4) found by the evolutionary algorithm on the training set of applications and evaluated them on the target applications and second, we let the evolutionary algorithm continue simulator-driven optimization on the target applications. The latter is unfair, in that the online approach is allowed access to querying more designs in the simulator. Nevertheless, we found that in either configuration, the evolutionary approach performed worse than PRIME which does not access training data from the target application domain. For the area constraint 29 mm$^2$ and 100 mm$^2$, the Evolutionary algorithm reduces the latency from 1127.64 → 820.11 and 861.69 → 552.64, respectively, although still worse than PRIME. In the second experiment in which we *unfairly* allow the evolutionary algorithm to continue optimizing on the target application, the Evolutionary algorithm suggests worse designs than Table 5 (e.g. 29 mm$^2$: 1127.64 → 1181.66 and 100 mm$^2$: 861.69 → 861.66).

**Table 13:** Optimized objective values (i.e., latency in milliseconds) obtained by various methods for the task of learning accelerators specialized to a given application. Lower latency/runtime is better. From left to right: our method, our method without negative sampling ("PRIME-Opt") and without utilizing infeasible points ("PRIME-Infeasible"), standard surrogate ("Standard"), online Bayesian optimization ("Bayes Opt"), online evolutionary algorithm ("Evolutionary") and the best design in the training dataset. Note that, in all the applications PRIME improves over the best in the dataset, outperforms online optimization methods in 7/9 applications and the complete version of PRIME generally performs best. The best accelerator designs are in bold.

| Application | PRIME | | | Standard | Online Optimization | | $\mathcal{D}$ (Best in Training) |
| | All | -Opt | -Infeasible | | Bayes Opt | Evolutionary | |
|---|---|---|---|---|---|---|---|
| MobileNetEdge | **298.50** | 435.40 | 322.20 | 411.12 | 319.00 | 320.28 | 354.13 |
| MobileNetV2 | **207.43** | 281.01 | 214.71 | 374.52 | 240.56 | 238.58 | 410.83 |
| MobileNetV3 | **454.30** | 489.45 | 483.96 | 575.75 | 534.15 | 501.27 | 938.41 |
| M4 | **370.45** | 478.32 | 432.78 | 1139.76 | 396.36 | 383.58 | 779.98 |
| M5 | 208.21 | 319.61 | 246.80 | 307.57 | 201.59 | **198.86** | 449.38 |
| M6 | 131.46 | 197.70 | 162.12 | 180.24 | 121.83 | **120.49** | 369.85 |
| U-Net | **740.27** | 740.27 | 765.59 | 763.10 | 872.23 | 791.64 | 1333.18 |
| t-RNN Dec | **132.88** | 172.06 | 135.47 | 136.20 | 771.11 | 770.93 | 890.22 |
| t-RNN Enc | **130.67** | 134.84 | 137.28 | 150.21 | 865.07 | 865.07 | 584.70 |

**Table 14:** The comparison between the accelerator designs suggested by PRIME and EdgeTPU [65, 23] for single model specialization. On average (last row), with single-model specialization our method reduces the latency by $2.69\times$ while minimizes the chip area usage by $1.50\times$.

| Application | Latency (milliseconds) | | | Chip Area (mm$^2$) | | |
| | PRIME | EdgeTPU | Improvement | PRIME | EdgeTPU | Improvement |
|---|---|---|---|---|---|---|
| MobileNetEdgeTPU | 294.34 | 523.48 | $1.78\times$ | 18.03 | 27 | $1.50\times$ |
| MobileNetV2 | 208.72 | 408.24 | $1.96\times$ | 17.11 | 27 | $1.58\times$ |
| MobileNetV3 | 459.59 | 831.80 | $1.81\times$ | 11.86 | 27 | $2.28\times$ |
| M4 | 370.45 | 675.53 | $1.82\times$ | 19.12 | 27 | $1.41\times$ |
| M5 | 208.42 | 377.32 | $1.81\times$ | 22.84 | 27 | $1.18\times$ |
| M6 | 132.98 | 234.88 | $1.77\times$ | 16.93 | 27 | $1.59\times$ |
| U-Net | 1465.70 | 2409.73 | $1.64\times$ | 25.27 | 27 | $1.07\times$ |
| t-RNN Dec | 132.43 | 1384.44 | $10.45\times$ | 14.82 | 27 | $1.82\times$ |
| t-RNN Enc | 130.45 | 1545.07 | $11.84\times$ | 19.87 | 27 | $1.36\times$ |
| **Average Improvement** | — | — | **$2.69\times$** | — | — | **$1.50\times$** |

## A.6 PRIME ABLATION STUDY

Here we ablate over variants of our method: (1) Opt was not used for negative sampling ("PRIME-Opt" in Table 13) (2) infeasible points were not used ("PRIME-Infeasible" in Table 13). As shown in Table 13, the variants of our method generally performs worse compared to the case when both negative sampling and infeasible data points are utilized in training the surrogate model.

## A.7 COMPARISON WITH HUMAN-ENGINEERED ACCELERATORS

In this section, we compare the optimized accelerator design found by PRIME that is targeted towards single applications to the manually optimized EdgeTPU design [65, 23]. EdgeTPU accelerators are primarily optimized towards running applications in image classification, particularly, MobileNetV2, MobileNetV3 and MobileNetEdgeTPU. The goal of this comparison is to present the potential benefit of PRIMEfor a dedicated application when compared to human designs. For this comparison, we utilize an area constraint of 27 mm$^2$ and a DRAM bandwidth of 25 Gbps, to match the specifications of the EdgeTPU accelerator.

Table 14 shows the summary of results in two sections, namely "Latency" and "Chip Area". The first and second under each section show the results for PRIME and EdgeTPU, respectively. The final column for each section shows the improvement of the design suggested by PRIME over EdgeTPU. On average (as shown in the last row), PRIME finds accelerator designs that are $2.69\times$ (up to $11.84\times$ in t-RNN Enc) better than EdgeTPU in terms of latency. Our method achieves this improvement while, on average, reducing the chip area usage by $1.50\times$ (up to $2.28\times$ in MobileNetV3). Even on the MobileNet image-classification domains, we attain an average improvement of $1.85\times$.

**Table 15:** Optimized objective values (i.e., latency in milliseconds) under zero-shot setting when the test applications include all the nine evaluated models (e.g. MobileNet (EdgeTPU, V2, V3), M4, M5, M6, t-RNN Dec, t-RNN Enc, U-Net). Lower latency is better. From **left** to **right**: the applications used to train the surrogate model in PRIME the target applications for which the accelerator is optimized for, the area constraint of the accelerator, PRIME's (best, median) latency, and best online method's (best, median) latency. The best accelerator configurations identified is highlighted in bold.

| Train Applications | Area | PRIME | Evolutionary (Online) |
|---|---|---|---|
| MobileNet (EdgeTPU,V2,V3), M4, M5, M6, t-RNN Enc | 29 mm$^2$ | (**426.65**, **427.94**) | (586.55, 586.55) |
| MobileNet (EdgeTPU,V2,V3), M4, M5, M6, t-RNN Enc | 100 mm$^2$ | (**365.95**, **366.64**) | (518.58, 519.37) |
| **Geomean of PRIME's Improvement** | — | (1.0×, 1.0×) | (**1.40×, 1.39×**) |

## A.8 ZERO-SHOT RESULTS ON ALL APPLICATIONS

In this section, we present the results of zero-shot optimization from Table 5 on all the nine applications we study in the paper (i.e., test applications = all nine models: MobileNet (EdgeTPU, V2, V3), M6, M5, M4, t-RNN (Enc and Dec), and U-Net). We investigate this for two sets of training applications and two different area budgets. As shown in Table 15, we find that PRIME does perform well compared to the online evolutionary method.

## A.9 DIFFERENT TRAIN AND VALIDATION SPLITS

In the main paper, we used the worst 80% of the feasible points in the training dataset for training and used the remaining 20% of the points for cross-validation using our strategy based on Kendall's rank correlation. In this section, we explore some alternative training-validation split strategies to see how they impact the results. To do so, we consider two alternative strategies: **(1)** training on 95% of the worst designs, validation on top 5% of the designs, and **(2)** training on the top 80% of the designs and validation on the worst 20% of the designs. We apply these strategies to MobileNetEdgeTPU, M6 and t-RNN Enc models from Table 3, and present a comparative evaluation in Table 16 below.

**Results.** As shown in Table 16, we find that cross-validating using the best 5% of the points in the dataset led to a reduced latency (298.50 → 273.30) on MobileNetEdgeTPU, and retained the same performance on M6. However, it increased the latency on t-RNN Enc (130.67 → 137.45). This indicates at the possibility that while top 5% of the datapoints can provide a better signal for cross-validation in some cases, this might also hurt performance if the size of the 5% dataset becomes extremely small (as in the case of t-RNN Enc, the total dataset size is much smaller than either MobileNetEdgeTPU or M6).

The strategy of cross-validating using the worst 20% of the points hurt performance on M6 and t-RNN Enc, which is perhaps as expected, since the worst 20% of the points may not be indicative of the best points found during optimization. However, while it improves performance on the MobileNetEdgeTPU application compared to the split used in the main paper but it is still worse than using the top 5% of the points for validation.

**Table 16:** Performance of PRIME (as measured by median latency of the accelerator found across five runs) under various train-test splits on three applications studied in Table 3.

| Applications | Best 5% Validation | Best 20% Validation (Table 3) | Worst 20% Validation |
|---|---|---|---|
| MobileNetEdgeTPU | 273.30 | 298.50 | 286.53 |
| M6 | 131.46 | 131.46 | 142.68 |
| t-RNN Enc | 137.45 | 130.67 | 135.71 |

## B DETAILS OF PRIME

In this section, we provide training details of our method PRIME including hyperparameters and compute requirements and details of different tasks.

## B.1 HYPERPARAMETER AND TRAINING DETAILS

Algorithm 1 outlines our overall system for accelerator design. PRIME parameterizes the function $f_\theta(\mathbf{x})$ as a deep neural network as shown in Figure 4. The architecture of $f_\theta(\mathbf{x})$ first embeds the

discrete-valued accelerator configuration $\mathbf{x}$ into a continuous-valued 640-dimensional embedding via two layers of a self-attention transformer [59]. Rather than directly converting this 640-dimensional embedding into a scalar output via a simple feed-forward network, which we found a bit unstable to train with Equation 3, possibly due to the presence of competing objectives for a comparison), we pass the 640-dimensional embedding into $M$ different networks that map it to $M$ different scalar predictions $(f_\theta^i(\mathbf{x}))_{i=1}^M$. Finally, akin to attention [59] and mixture of experts [52], we train an additional head to predict weights $(w_i)_{i=1}^M \geq 0$ of a linear combination of the predictions at different heads that would be equal to the final prediction: $f_\theta(\mathbf{x}) = \sum_{i=1}^K w_i f_\theta^i(\mathbf{x})$. Such an architecture allows the model to use different predictions $f_\theta^i(\mathbf{x})$, depending upon the input, which allows for more stable training. To train $f_\theta(\mathbf{x})$, we utilize the Adam [32] optimizer. Equation 3 utilizes a procedure Opt that maximizes the learned function approximately. We utilize the same technique as Section 4 ("optimizing the learned surrogate") to obtain these negative samples. We periodically refresh Opt, once in every 20K gradient steps on $f_\theta(\mathbf{x})$ over training.

---

**Algorithm 1** Training the conservative surrogate in PRIME

1: Initialize a neural model $f_{\theta_0}(\mathbf{x})$ and a set $M = 23$ of negative particles to be updated by the firefly optimizer $\{\mathbf{x}_1^-(0), \cdots, \mathbf{x}_i^-(0), \mathbf{x}_M^-(0)\}$ to random configurations from the design space.
2: **for** iteration $i = 0, 1, 2, 3, \ldots$ until convergence **do**
3:     **for** firefly update step $t = 0, 1, \ldots, 4$                               ▷ **Inner loop do**
4:         Update the $M$ fireflies according to the firefly update rule in Equation 5,
5:         towards maximizing $f_{\theta_i}(\mathbf{x})$ according to:            ▷ **Negative mining**
6:                $\mathbf{x}_i(ti+1) = \mathbf{x}_i(ti) + \beta(\mathbf{x}_i(ti) - \mathbf{x}_j(ti)) + \eta\epsilon_{ti}$
7:     **end for**
8:     Find the best firefly found in these steps to be used as the negative sample:
9:         $\mathbf{x}_i^- = \arg\min\{f_{\theta_i}(\mathbf{x}_1^-(ti)), \cdots, f_{\theta_i}(\mathbf{x}_M^-(ti))\}$     ▷ **Find negative sample**
10:     Run one gradient step on $\theta_i$ using Equation 3 with $\mathbf{x}_i^-$ as the negative sample
11:     **if** $i\%p == 0$, (p = 20000), then:        ▷ **Periodically reinitialize the optimizer**
12:         Reinitialize firefly particles $\{\mathbf{x}_1^-(0), \cdots, \mathbf{x}_i^-(0), \mathbf{x}_M^-(0)\}$ to random designs.
13: **end for**
14: Return the final model $f_{\theta^*}(\mathbf{x})$

---

The hyperparameters for training the conservative surrogate in Equations 3 and its contextual version are as follows:

- **Architecture of $f_\theta(\mathbf{x})$.** As indicated in Figure 4, our architecture takes in list of categorical (one-hot) values of different accelerator parameters (listed in Table 1), converts each parameter into 64-dimensional embedding, thus obtaining a $10 \times 64$ sized matrix for each accelerator, and then runs two layers of self-attention [59] on it. The resulting $10 \times 64$ output is flattened to a vector in $\mathbb{R}^{640}$ and fed into $M = 7$ different prediction networks that give rise to $f_\theta^1(\mathbf{x}), \cdots, f_\theta^M(\mathbf{x})$, and an additional attention 2-layer feed-forward network (layer sizes $= [256, 256]$) that determines weights $w_1, \cdots, w_M$, such that $w_i \geq 0$ and $\sum_{i=1}^M w_i = 1$. Finally the output is simply $f_\theta(\mathbf{x}) = \sum_i w_i f_\theta^i(\mathbf{x})$.

- **Optimizer/learning rate for training $f_\theta(\mathbf{x})$.** Adam, $1e-4$, default $\beta_1 = 0.9, \beta_2 = 0.999$.

- **Validation set split.** Top 20% high scoring points in the training dataset are used to provide a validation set for deciding coefficients $\alpha$, $\beta$ and the checkpoint to evaluate.

- **Ranges of $\alpha$, $\beta$.** We trained several $f_\theta(\mathbf{x})$ models with $\alpha \in [0.0, 0.01, 0.1, 0.5, 1.0, 5.0]$ and $\beta \in [0.0, 0.01, 5.0, 0.1, 1.0]$. Then we selected the best values of $\alpha$ and $\beta$ based on the highest Kendall's ranking correlation on the validation set. Kendall's ranking correlation between two sets of objective values: $S = \{y_1, y_2, \cdots, y_N\}$ corresponding to ground truth latency values on the validation set and $S' = \{y_1', y_2', \cdots, y_N'\}$ corresponding to the predicted latency values on the validation set is given by $\tau$ equal to:

$$\tau = \frac{\sum_{i,j}^{N,N} \mathbb{I}[(y_i - y_j)(y_i' - y_j') > 0] - \sum_{i,j}^{N,N} \mathbb{I}[(y_i - y_j)(y_i' - y_j') \leq 0]}{N \cdot (N-1)}. \quad (4)$$

**Table 17:** Dataset sizes for various applications that we study in this paper. Observe that all of the datasets are smaller than 8000.

| Application | Dataset size |
|---|---|
| MobileNetEdgeTPU | 7697 |
| MobileNetV2 | 7620 |
| MobileNetV3 | 5687 |
| M4 | 3763 |
| M5 | 5735 |
| M6 | 7529 |
| U-Net | 557 |
| t-RNN Dec | 1211 |
| t-RNN Enc | 1240 |

- **Clipping $f_\theta(\mathbf{x})$ during training**. Equation 3 increases the value of the learned function $f_\theta(\mathbf{x})$ at $\mathbf{x} = \mathbf{x}_0 \in \mathcal{D}_{\text{infeasible}}$ and $\mathbf{x}^- \sim \text{Opt}(f_\theta)$. We found that with the small dataset, these linear objectives can run into numerical instability, and produce $+\infty$ predictions. To avoid this, we clip the predicted function value both above and below by $\pm 10000.0$, where the valid range of ground-truth values is $\mathcal{O}(1000)$.

- **Negative sampling with $\text{Opt}(\cdot)$.** As discussed in Section 4, we utilize the firefly optimizer for both the negative sampling step and the final optimization of the learned conservative surrogate. When used during negative sampling, we refresh (i.e., reinitialize) the firefly parameters after every $p = 20000$ gradient steps of training the conservative surrogate, and run $t = 5$ steps of firefly optimization per gradient step taken on the conservative surrogate.

- **Details of firefly:** The initial population of fireflies depends on the number of accelerator configurations ($\mathcal{C}$) following the formula $10 + \lfloor (\mathcal{C}^{1.2} + \mathcal{C}) \times 0.5 \rceil$. In our setting with ten accelerator parameters (See Table 1), the initial population of fireflies is 23. We use the same hyperparameters: $\gamma = 1.0, \beta_0 = 1.0$, for the optimizer in all the experiments and never modify it. The update to a particular optimization particle (i.e., a firefly) $\mathbf{x}_i$, at the $t$-th step of optimization is given by:

$$\mathbf{x}_i(t+1) = \mathbf{x}_i(t) + \beta(\mathbf{x}_i(t) - \mathbf{x}_j(t)) + \text{ i.i.d. Gaussian noise}, \tag{5}$$

  where $\mathbf{x}_j(t), j \neq i$ is a different firefly that achieves a better objective value compared to $\mathbf{x}_i$ and the function $\beta$ is given by: $\beta(r) = \beta_0 e^{-\gamma r^2}$.

- **Training set details:** The training dataset sizes for the studied applications are shown in Table 17. To recap, to generate the dataset, we first randomly sampled accelerators from the deign space, and evaluated them for the target application, and constituted the training set from the worst-performing feasible accelerators for the given application. Since different applications admit different feasibility criteria (differences in compilation, hardware realization, and etc.), the dataset sizes for each application are different, as the number of feasible points is different. Note however that as mentioned in the main text, these datasets all contain $\leq 8000$ feasible points.

  **Discussion on data quality:** In the cases of t-RNN Dec, t-RNN Enc, and U-Net, we find that the number of feasible points is much smaller compared to other applications, and we suspect this is because our random sampling procedure does not find enough feasible points. This is a limitation of our data collection strategy and we intentionally chose this naïve strategy to keep data collection simple. Other techniques for improving data collection and making sure that the data does not consist of only infeasible points includes strategies such as utilizing logged data from past runs of online evolutionary methods, mixed with some data collected via random sampling to improve coverage of the design space.

**Table 18:** Comparing the latency of the accelerators designed by the evolutionary approach for variable number of simulator access budgets (8k and 32k). Even with $4\times$ as much allowed simulator interaction, online methods are unable to perform that well in our case.

| Application | Area | PRIME | Evolutionary (Online) 8k data points | Evolutionary (Online) 32k data points |
|---|---|---|---|---|
| MobileNetEdgeTPU | 29 mm$^2$ | 298.50 | 320.28 | 311.35 |
| t-RNN Dec | 29 mm$^2$ | 132.88 | 770.93 | 770.63 |
| t-RNN Enc | 29 mm$^2$ | 130.67 | 865.07 | 865.07 |
| **Geomean of PRIME's Improvement** | — | $(1.0\times, 1.0\times)$ | **3.45$\times$** | **3.42$\times$** |

### B.1.1 DETAILS OF FIREFLY USED FOR OUR ONLINE EVOLUTIONARY METHOD

In this section, we discuss some details for firefly optimization used in the online evolutionary method.

**Stopping criterion:** We stopped the firefly optimization when the latency of the best design found did not improve over the previous 1000 iterations, but we also made sure to run firefly optimization for at least 8000 iterations, to make sure that both the online and offline methods match in terms of the data budget. We also provide the convergence curves for firefly optimization on various single-application problems from Table 3 in Figure 8.

**What happens if we run firefly optimization for longer?** We also experimented with running the evolutionary methods for longer (i.e., 32k simulator accesses compared to 8k), to check if this improves the performance of the evolutionary approach. As shown in Table 18, we find that while this procedure does improve performance in some cases, the performance does not improve much beyond 8k steps. This indicates that there is a possibility that online methods can perform better than PRIME if they are run for many more optimization iterations against the simulator, but they may not be as data-efficient as PRIME.

**Hyperparameter tuning for firefly:** Since the online optimization algorithms we run have access to querying the simulator over the course of training, we can simply utilize the value of the latest proposed design as a way to perform early stopping and hyperparameter tuning. A naïve way to perform hyperparameter tuning for such evolutionary methods is to run the algorithm for multiple rounds with multiple hyperparameters, however this is compute and time intensive. Therefore, we adopted a dynamic hyperparameter tuning strategy. Our implementation of the firefly optimizer tunes hyperparameters by scoring a set of hyperparameters based on its best performance over a sliding window of $T$ data points. This allows us to adapt to the best hyperparameters on the fly, within the course of optimization, effectively balancing the number of runs that need to be run in the simulator and hyperparameter tuning. This dynamic hyperparameter tuning strategy requires some initial coverage of the hyperparameter space before hyperparameter tuning begins, and therefore, this tuning begins only after 750 datapoints. After this initial phase, every $T = 50$ iterations, the parameters $\gamma$ and $\beta_0$ are updated via an evolutionary scoring strategy towards their best value.

**Discussion of t-RNN Enc and t-RNN Dec**. Finally, we discuss the results of the evolutionary approach on the t-RNN Enc and t-RNN Dec tasks, for which the convergence plots are shown in Figures 8h and 8i. Observe that the best solution found by this optimization procedure converges quite quickly in this case (with about 1000 iterations) and the evolutionary method, despite the dynamic hyperparameter tuning is unable to find a better solution. We hypothesize that this is because the performance of a local optimization method may suffer heavily due to the poor landscape of the objective function, and it may get stuck if it continuously observes only infeasible points over the course of optimization.

### B.1.2 EXACT HYPERPARAMETERS FOUND BY OUR CROSS-VALIDATION STRATEGY

In this section, we present the exact hyperparameters found by our cross-validation strategy discussed in Section 4. To recap, our offline cross-validation strategy finds the early stopping checkpoint and selects the values of $\alpha$ and $\beta$ in Equation 3 that attain the highest rank correlation on a held-out validation set consisting of top 20% of the dataset feasible samples. The values of $\alpha$, $\beta$ and checkpoint selected for the experiments in Table 3, Table 4, Table 5 and 6 are shown in Table 19.

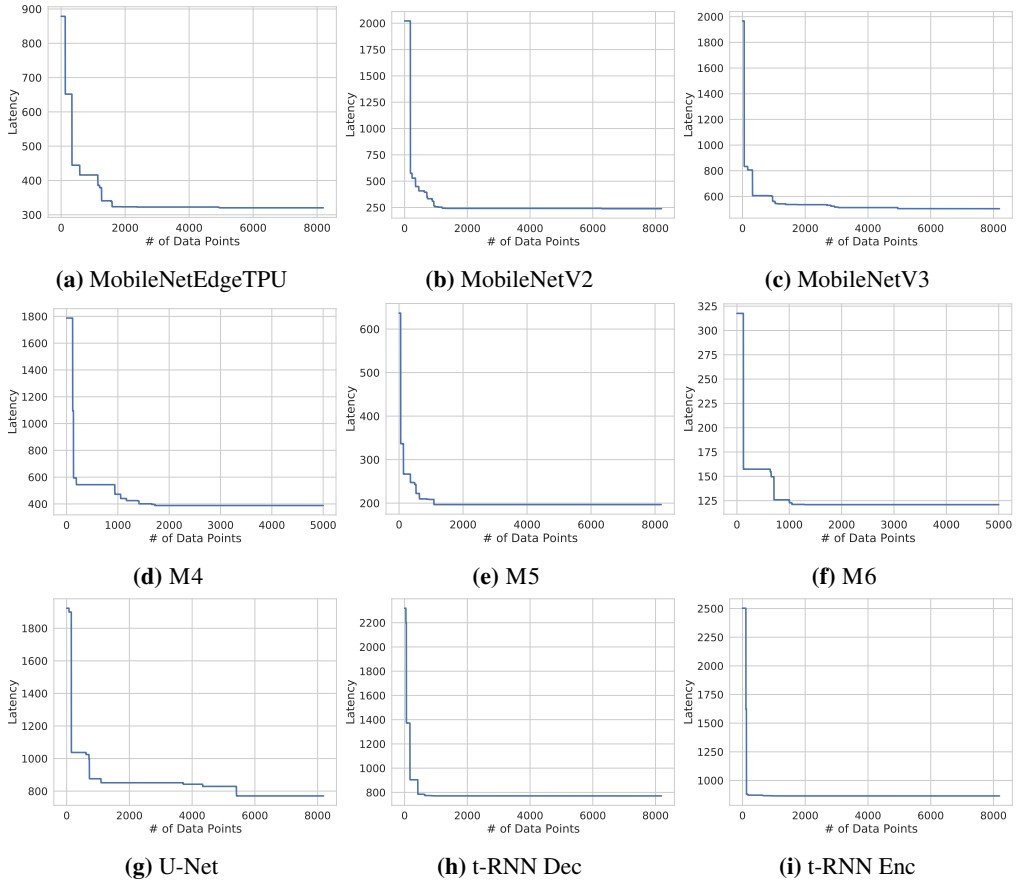

**Figure 8:** Optimization behavior of Firefly optimizer (Online Evolutionary). Observe that the optimization procedure converges and plateaus very quickly (at least 1000 iterations in advance) and hence we stop at 8000 iterations. In the case of t-RNN Enc and t-RNN Dec, we find that the evolutionary algorithm performs poorly and we suspect this is because it saturates quite quickly to a suboptimal solution and is unable to escape. This is also evident from Figures 8h and 8i, where we observe that online optimization plateaus the fastest for these RNN applications.

## B.2 DETAILS OF ARCHITECTING ACCELERATORS FOR MULTIPLE APPLICATIONS SIMULTANEOUSLY

Now we will provide details of the tasks from Table 4 where the goal is to architect an accelerator which is jointly optimized for multiple application models. For such tasks, we augment data-points for each model with the context vector $c_k$ from Table 2 that summarizes certain parameters for each application. For entries in this context vector that have extremely high magnitudes (e.g., model parameters and number of compute operations), we normalize the values by the sum of values across the applications considered to only encode the relative scale, and not the absolute value which is not required. To better visualize the number of feasible accelerators for joint optimization, Figure 9 show the tSNE plot (raw architecture configurations are used as input) of high-performing accelerator configurations. The blue-colored dots are the jointly feasible accelerators in the combined dataset, and note that these data points are no more than 20-30 in total. The highlighted red star presents the best design suggested by PRIME with average latency of 334.70 (Table 4). This indicates that this contextual, multi-application problem poses a challenge for data-driven methods: these methods need to produce optimized designs even though very few accelerators are jointly feasible in the combined dataset. Despite this limitation, PRIME successfully finds more efficient accelerator configurations that attain low latency values on each of the applications jointly, as shown in Table 4.

**Table 19:** Hyperparameters $\alpha$, $\beta$ and checkpoint index (measured in terms of gradient steps on the learned conservative model) for PRIME found by our **offline** cross-validation strategy discussed in Section 4, that is based on the Kendall's rank correlation on the validation set (note that no simulator queries were used to tune hyperparameters). In the case of the multi-task and zero-shot scenarios, when training on more than one application, the batch size used for training PRIME increases to $N$-fold, where $N$ is the number of applications in the training set, therefore we likely find that even a few gradient steps are good enough.

| Table | Application | $\alpha$ | $\beta$ | Checkpoint Index |
|---|---|---|---|---|
| Table 3 | MobileNetEdgeTPU | 0.01 | 5.0 | 80000 |
| Table 3 | MobileNetV2 | 5.0 | 5.0 | 120000 |
| Table 3 | MobileNetV3 | 5.0 | 0.01 | 80000 |
| Table 3 | M4 | 0.1 | 0.0 | 80000 |
| Table 3 | M5 | 5.0 | 1.0 | 80000 |
| Table 3 | M6 | 1.0 | 1.0 | 60000 |
| Table 3 | U-Net | 0.0 | 1.0 | 100000 |
| Table 3 | t-RNN Dec | 1.0 | 0.0 | 60000 |
| Table 3 | t-RNN Enc | 0.0 | 0.1 | 60000 |
| Table 4 | MobileNet (EdgeTPU, V2, V3) | 5.0 | 0.01 | 60000 |
| Table 4 | MobileNet (V2, V3), M5, M6 | 0.0 | 5.0 | 30000 |
| Table 4 | MobileNet (EdgeTPU, V2, V3), M4, M5, M6 | 0.5 | 0.0 | 100000 |
| Table 4 | MobileNet (EdgeTPU, V2, V3), M4, M5, M6, t-RNN Enc (Area 29.0) | 0.0 | 1.0 | 20000 |
| Table 4 | MobileNet (EdgeTPU, V2, V3), M4, M5, M6, t-RNN Enc (Area 100.0) | 0.0 | 0.0 | 20000 |
| Table 4 | MobileNet (EdgeTPU, V2, V3), M4, M5, M6, U-Net, t-RNN (Enc, Dec) (Area 29.0) | 0.01 | 0.01 | 10000 |
| Table 4 | MobileNet (EdgeTPU, V2, V3), M4, M5, M6, U-Net, t-RNN (Enc, Dec) (Area 100.0) | 0.01 | 0.1 | 20000 |
| Table 5 | Train (Zero-Shot): MobileNet (EdgeTPU, V3) | 5.0 | 0.01 | 60000 |
| Table 5 | Train (Zero-Shot): MobileNet (V2, V3), M5, M6 | 0.0 | 5.0 | 30000 |
| Table 5 | Train (Zero-Shot): MobileNet (EdgeTPU, V2, V3), M4, M5, M6, t-RNN Enc (Area 29.0) | 0.0 | 1.0 | 20000 |
| Table 5 | Train (Zero-Shot): MobileNet (EdgeTPU, V2, V3), M4, M5, M6, t-RNN Enc (Area 100.0) | 0.1 | 5.0 | 20000 |
| Table 6 | MobileNetV2 (NVDLA) | 0.0 | 1.0 | 40000 |
| Table 6 | MobileNetV2 (ShinDianNao) | 0.0 | 0.0 | 40000 |
| Table 6 | ResNet 50 (NVDLA) | 0.01 | 0.0 | 40000 |
| Table 6 | ResNet 50 (ShinDianNao) | 0.0 | 0.0 | 75000 |
| Table 6 | Transformer (NVDLA) | 0.01 | 1.0 | 200000 |
| Table 6 | Transformer (ShinDianNao) | 0.0 | 0.1 | 100000 |

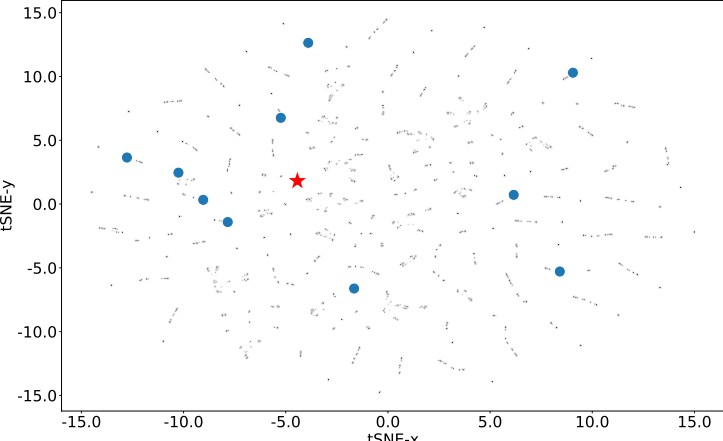

**Figure 9:** tSNE plot of the joint dataset and randomly sampled infeasible data points. The blue points show the accelerator configurations that are jointly feasible for all the applications. The highlighted point with red star shows the best design proposed by PRIME. The rest of the points show the infeasible points.

### B.3  DATASET SENSITIVITY TO ACCELERATOR PARAMETERS

We visualize the sensitivity of the objective function (e.g. latency) with respect to the changes in certain accelerator parameters, such as memory size (Table 1), in Figure 11b, illustrating this sensitivity. As shown in the Figure, the latency objective that we seek to optimize can exhibit high sensitivity to small variations in the architecture parameters, making the optimization landscape particularly ill-behaved. Thus, a small change in one of the discrete parameters, can induce a large change in the optimization objective. This characteristic of the dataset further makes the optimization task challenging.

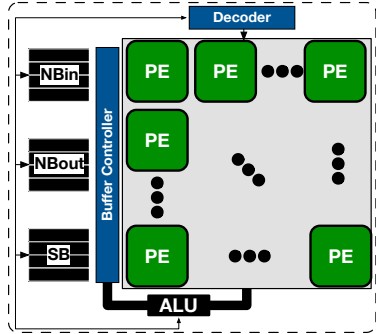

**Figure 10:** Overview of ShiDianNao dataflow accelerator. This dataflow accelerators exhibits an output-stationary dataflow where it keeps the partial results stationary within each processing elements (PEs).

**Table 20:** The evaluated applications, their model parameter size, number of compute operations, and normalized compute-to-memory ratio.

| Name | Model Param | # of Compute Ops. | Normalized Compute-to-Memory Ratio |
|---|---|---|---|
| **MobileNetEdgeTPU** | 3.87 MB | 1,989,811,168 | 1.38e-1 |
| **MobileNetV2** | 3.31 MB | 609,353,376 | 4.96e-2 |
| **MobileNetV3** | 5.20 MB | 449,219,600 | 2.33e-2 |
| **M4** | 6.23 MB | 3,471,920,128 | 1.5e-1 |
| **M5** | 2.16 MB | 939,752,960 | 1.17e-1 |
| **M6** | 0.41 MB | 228,146,848 | 1.5e-1 |
| **U-Net** | 3.69 MB | 13,707,214,848 | 1.0 |
| **t-RNN Dec** | 19 MB | 40,116,224 | **5.68e-4** |
| **t-RNN Enc** | 21.62 MB | 45,621,248 | **5.68e-4** |

## C  OVERVIEW OF ACCELERATORS AND SEARCH SPACE

This section briefly discuss the additional accelerators (similar to [30]) that we evaluate in this work, namely NVDLA [43] and ShiDianNao [10], and their corresponding search spaces.

**NVDLA: Nvidia Deep Learning Accelerator** NVDLA [42] is an open architecture inference accelerator designed and maintained by Nvidia. In compared to other inference accelerators, NVDLA is a weight stationary accelerator. That is, it retains the model parameters on each processing elements and parallelizes the computations across input and output channels. NVDLA-style dataflow accelerators generally yield better performance for the computations of layers at the later processing stages. This is because these layers generally have larger model parameters that could benefit from less data movement associated to the model parameters.

**ShiDianNao: Vision Accelerator** Figure 10 shows the high-level schematic of ShiDianNao accelerator [10]. ShiDianNao-style dataflow accelerator is an output-stationary accelerator. That is, it keeps the partial results inside each PE and instead move the model parameters and input channel data. As such, in compared to NVDLA-style accelerators, ShiDianNao provides better performance for the computations of the layers with large output channels (generally first few layers of a model).

**Search space of dataflow accelerators.** We follow a similar methodology as [30] to evaluate additional hardware accelerators, discussed in the previous paragraphs. We use MAESTRO [37], an analytical cost model, that supports the performance modeling of various dataflow accelerators. In this joint accelerator design and dataflow optimization problem, the total number of parameters to be optimized is up to 106—the tuple of (# of PEs, Buffers) per per model layer—with each parameter taking one of 12 discrete values. This makes the hardware search space consist of $\approx 2.5 \times 10^{114}$ accelerator configurations. We also note that while the method proposed in [30] treats the accelerator design problem as a sequential decision making problem, and uses reinforcement learning techniques, PRIME simply designs the whole accelerator in a single step, treating it as a model-based optimization problem.

## D  SUBSET OF APPLICATIONS FOR GOOD ZERO-SHOT PERFORMANCE

In this section, we present the results of an ablation study with the goal to identify a subset of applications such that training on data from only these applications yields good zero-shot performance

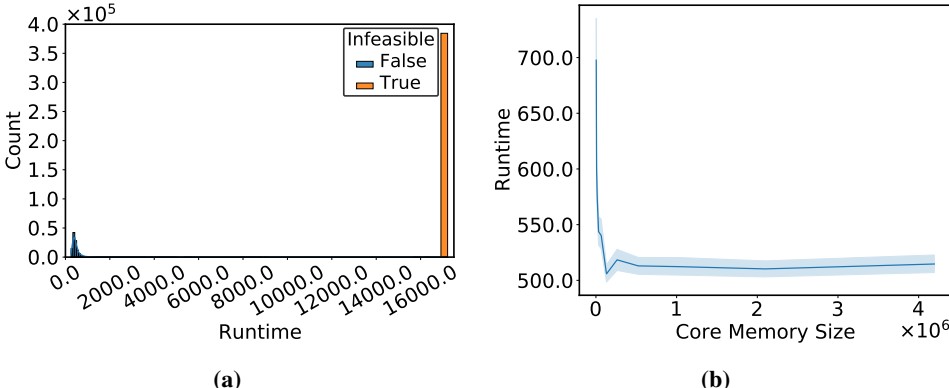

**Figure 11:** The (a) histogram of infeasible (orange bar with large score values)/feasible (blue bars) data points and (b) the sensitivity of runtime to the size of core memory for the MobileNetEdgeTPU [26] dataset.

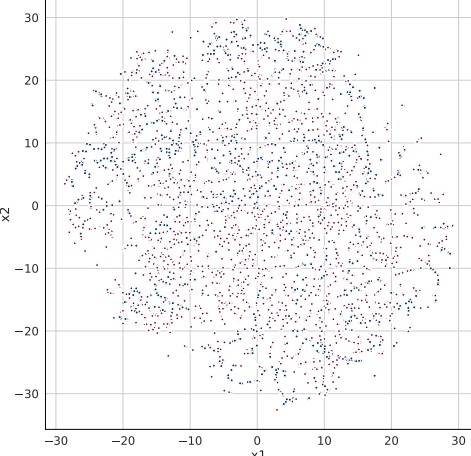

**Figure 12:** tSNE plot of the infeasible and feasible hardware accelerator designs. Note that feasible designs (shown in blue) are embedded in a sea of infeasible designs (shown in red), which makes this a challenging domain for optimization methods.

across all the nine applications studied in this work. Since we cannot train PRIME for every subset of applications because the space of subsets of all applications is exponentially large, we utilized some heuristics in devising the subset of applications we would train on, with the goal to make interesting observations that allow us to devise rough guidelines for performing application selection.

**Our heuristic for devising subsets of applications:** Building on the intuition that applications with very different compute to memory ratios (shown in Table 20) may require different accelerator designs – for example, if our goal is to run a compute-intensive application, we likely need an accelerator design with more compute units – we study two subsets of training applications: **(1)** MobileNetV2, MobileNetV3, M6, M5, and **(2)** MobileNetV2, MobilenetV3, M5, t-RNN Enc. Note that, these two combinations only differ in whether some RNN application was used in training or not. As shown in Table 20, the t-RNN applications admit a very different compute to memory ratio, for instance, while this ratio is $5.68e - 4$ for t-RNN Enc and t-RNN Dec, it is much different $\sim 0.01 - 0.2$ for other models MobileNetEdgeTPU, MobileNetV2, MobileNetV3, M5, and M6. This means that likely t-RNN Enc and Dec will require different kinds of accelerators for good performance compared to the other applications.

**Results:** We present the performance of zero-shot evaluating the designed accelerator obtained by training on combinations **(1)** and **(2)**, and also the accelerator obtained by training on **(3)** seven applications from Table 5 in Table 21 as reference. We make some key takeaways from the results:

- The performance of both configuration **(1)** and training with seven applications (**(3)**, last row of Table 21) are similar.

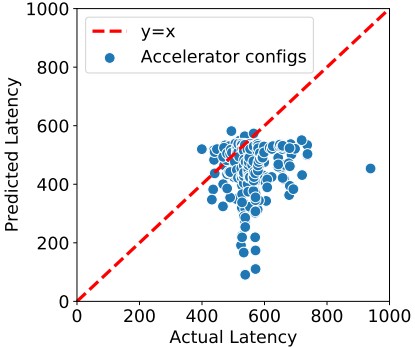

**Figure 13:** To verify if the overestimation hypothesis–that optimizing an accelerator against a naïve standard surrogate model is likely to find optimizers that appear promising under the learned model, but do not actually attain low-latency values–in our domain, we plot a calibration plot of the top accelerator designs found by optimizing a naïvely trained standard surrogate model. In the scatter plot, we represent each accelerator as a point with its x-coordinate equal to the actual latency obtained by running this accelerator design in the simulator and the y-coordinate equal to the predicted latency under the learned surrogate. Note that for a large chunk of designs, the predicted latency is much smaller than their actual latency (i.e., these designs lie beneath the $y = x$ line in the plot above). This means that optimizing designs under a naïve surrogate model is prone to finding designs that appear overly promising (i.e., attain lower predicted latency values), but are not actually promising. This confirms the presence of the overestimation hypothesis on our problem domain.

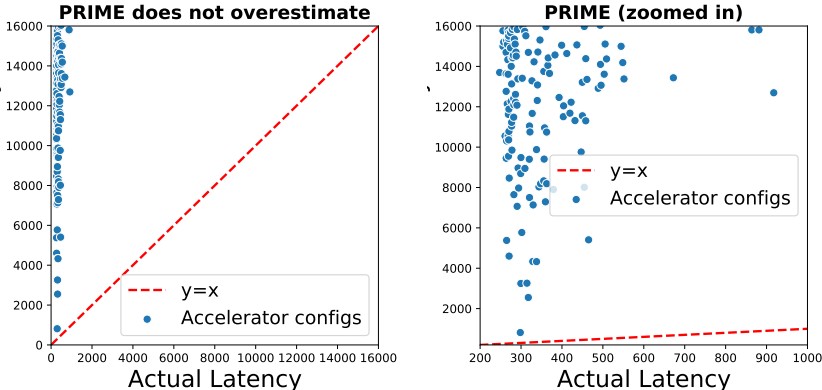

**Figure 14:** Plot showing the calibration plot of the predicted (y-axis) and actual latencies (x-axis) of accelerators found by PRIME. Compared to Figure 13, observe that all the acclerator configurations lie above $y = x$, meaning that PRIME predicts a higher latency (y-axis) compared to the actual latency. This means that PRIME does not think that accelerators that attain high-latency values under the simulator, are actually good. We also provide a zoomed-in version of the plot on the right, which shows that there are accelerators do have meaningfully distinct latency predictions under PRIME. Observe in the zoomed-in plot that the designs that attain small predicted latencies also perform relatively better under the actual latency compared to the designs that attain larger predicted latency of $\sim$ 14000-16000 under the PRIME surrogate. Optimizing against PRIME is still effective because optimization just needs relative correctness of values, not absolutely correct latency predictions.

- In case **(2)**, when the training applications consist of four applications in which one application is t-RNN Enc, with drastically different compute to memory ratio (Table 20), the performance on an average across all applications becomes slightly worse (compare the performance in **(2)** vs **(3)**).

**Conclusion and guidance on selecting good applications:** The above results indicate that only a few applications (e.g., four applications in case **(1)**) can be enough for good performance on all nine applications. While this set may not be not minimal, it is certainly a much smaller set compared to the nine applications considered. Adding an RNN application in case **(2)** increases latency in compared to case **(1)**, because t-RNN Enc likely admits a very different optimal accelerator compared to the other applications due to a very different compute/memory ratio, which in turn skews the generalization of the surrogate learned by PRIME when trained only on this limited set of four applications. However,

**Table 21:** Additional ablation study under zero-shot setting when the test applications include all the nine evaluated models (e.g. MobileNet (EdgeTPU, V2, V3), M4, M5, M6, t-RNN Dec, t-RNN Enc, U-Net). Lower latency is better. From **left** to **right**: the applications used to train the surrogate model in PRIME, the area constraint of the accelerator, PRIME's (best, median) latency.

| Train Applications | Area | PRIME (best, median) |
|---|---|---|
| **(1)** MobileNet (V2, V3), M5, M6 | 29 mm$^2$ | (426.65, 427.35) |
| **(2)** MobileNet (V2, V3), M5, t-RNN Enc | 29 mm$^2$ | (461.79, 464.87) |
| **(3)** MobileNet (EdgeTPU, V2, V3), M4, M5, M6, t-RNN Enc | 29 mm$^2$ | (426.65, 427.94) |

when seven applications are provided in case **(3)**, even when the set of training applications includes t-RNN, its contribution on the PRIME surrogate is reduced since many other compute intensive applications are also provided in the training set and the resulting accelerator performs well.

**Practitioner guidance:** The primary practitioner guidance we can conclude here is that *the models used for training must be representative of the overall distribution of the target models that we want to zero-shot generalize to.* Since a number of our applications are compute intensive, we were able to simply utilize set **(1)** to attain good performance on all the applications. On the other hand, in case **(2)**, when the t-RNN Enc application was over-represented – while seven of nine applications we considered were primarily compute intensive, one out of four applications we used for training in case **(2)** were memory intensive – this hurt performance on the overall set. Therefore, we believe that ensuring that the training subset of applications is adequately aligned with the overall set of applications in terms of compute/memory ratio statistic is imperative for favorable zero-shot performance.

**For a practitioner deciding between zero-shot generalization and additional data collection**, it may make sense to test if the target application admits a similar value of the compute to memory ratio as an already existing application. If it does, then the practitioner might be able to utilize the zero-shot generalization, as is indicated with the good performance of case **(1)**, whereas if the compute/memory ratio is heavily different from any seen application, zero-shot generalization to the target application may be worse. Finally, making sure that the training applications adequately reflect the compute/memory ratio statistic for the overall target set is important.

