# OpenReview forum: "Data-Driven Offline Optimization for Architecting Hardware Accelerators"
_ICLR.cc/2022/Conference — ICLR 2022 Poster_

### Official Review · Reviewer_aghq · 2021-11-02

**Correctness:** 4
**Technical Novelty And Significance:** 3
**Empirical Novelty And Significance:** 3
**Recommendation:** 8
**Confidence:** 4

**Main Review:**

The authors take a clearly defined problem (design a neural net hardware accelerator), motivate it well (simulations are expensive; surrogates struggle with these specific problems in this domain), introduce a solution targeted at overcoming those specific issues, and then demonstrate that it generally works better than other accepted approaches.

Strengths:

- The evaluation results are compelling. While the absolute design point gains are modest, they must be understood in the context of dramatically lower computational time, which is a huge selling point by itself. I appreciated the various evaluation axes the authors have explored: single accelerator, shared (multi-target) accelerator, zero-shot, and applicability to multiple accelerator design paradigms.
- The organization is logical, and the writing is for the most part clear. I appreciate that writing papers on hardware accelerators in the ML community is sometimes difficult due to wildly varying levels of background in the audience. I feel the authors have struck a solid balance here: detail enough for those with a background, but helpfully reducing the discussion to just the important pieces.

Weaknesses:

- The paper presents PRIME as its solution, however the actual novel contributions are not particularly well analyzed. The authors highlight infeasible design points and a rough optimization landscape as the key problem characteristics, but the evaluation results and analysis are almost strictly black-box tests. This is a bit unfortunate because it makes it harder to understand how the proposed modifications are contributing to actually solving the described problems in a quantitative way. Some of this is discussed in appendices (e.g., the ablation study in A.6), but it is brief (and relegated to an appendix).

- The paper is squished. It seems clear the authors had a lot of material to incorporate, but quality has been sacrificed somewhat. Margins and labels on figures and tables in the paper are hard to read and run into text.



Disclosure: I have previously anonymously reviewed a version of this work. Comments below reflect the progress and are mostly for the authors' benefit.

I applaud the authors for the changes they have incorporated into this paper based on past reviews. Prior concerns about generalizability on both application and hardware are largely gone, and the paper has mostly retained its clarity. Some of the intuitive explanations have been omitted (probably due to space concerns), but it still generally holds up. This is a better paper.

I did notice that the discussion of prediction behavior as it scales with dataset sample size is gone. This is a bit unfortunate since it was one of the pieces that helped shed light on how the method worked (and where it breaks down). I don't believe this fundamentally harms the takeaway of the paper (since the existing evaluations are pretty clear that the method can be made to work in a variety of situations), but it might be nice if that analysis made its way back into an appendix.

**Summary Of The Paper:**

The authors present an offline performance model for optimizing neural net hardware accelerators. They introduce a mechanism for dealing with non-viable hardware design points and for specializing towards specific applications from a single surrogate. They present results versus online methods on a variety of axes.

**Summary Of The Review:**

The paper is clear, describes its problem and solution well, clearly showcases its results across a variety of tests, and generally succeeds at convincing the reader that the approach works as advertised. There is room for improving the 'why', but the paper is solid nonetheless.

---

> ### Author Response · Authors · 2021-11-20
> **Author Response**
>
> We thank the reviewer for their constructive feedback and for a positive assessment of the work. We are glad that the reviewer found the results compelling and that this submission is a better paper compared to the previous version.
>
> To address the reviewer’s comment, we are now editing the writing and the formatting a bit to make the content clear, and we will aim to finish it for the final version of the paper. We have attempted to make the figures not run into text margins. Per the reviewer’s request, we have added the ablation study discussing the overfitting/underfitting behavior in Appendix A.8.
>
> Regarding the “why” question, we agree that this is an important question to answer. To take an initial step to addressing this, we present an analysis of the solution found by PRIME  in Appendix A.4 and compare the optimized accelerator design to online methods. This analysis indicates that PRIME designs a very different accelerator, one that better trades off memory and compute size for the nine application case to accommodate the t-RNN Dec and t-RNN Enc. Specifically, since t-RNN Enc and t-RNN Dec require more memory, PRIME reduces the number of compute cores needed to accommodate the memory requirement. Of course, this does not fully answer the “why” question fully, we hope that this analysis provides some insights into the solutions found by PRIME.

---

### Official Review · Reviewer_cuvZ · 2021-11-03

**Correctness:** 3
**Technical Novelty And Significance:** 2
**Empirical Novelty And Significance:** Not applicable
**Recommendation:** 8
**Confidence:** 4

**Main Review:**

## Novelty and Significance

The paper addresses an important emerging problem, designing custom hardware architectures to accelerate machine learning algorithms; however, the presented algorithm is general to any configuration tuning / design space exploration approach. None of the techniques within the paper are deeply novel (using adversarial training in the loss function [1], adding feasibility as a term in the loss function [2], employing context vectors to reuse design space exploration for multiple applications [3], and the actual optimizer used to optimize against Prime [4]), the combination of approaches and application to the specific problem under study is sufficiently novel to be worth publishing. Regardless, I would appreciate a more strong statement of precisely what components of the paper should be considered novel.

## Correctness and Clarity

At a high level, the approach and evaluation in the paper seems mostly correct, and the paper is very easy to read. However, I do have the following questions and concerns:
- Many of the details of the evaluation are omitted. Specifically:
  - (*) what is the size of the training set?
  - (*) What is the stopping criteria for the various techniques (specifically, how many iterations is fireflies ran for against the surrogate)?
  - What is the total amount of compute for Prime? Figures 5 and 6 show the simulation time (which seems to be increasing for Prime, which I also don't understand, since I had thought that Prime was entirely offline), but does not include the cost of training and evaluating the surrogate.
- I am not entirely convinced that all of the experimental results are correct. Specifically, the following results are surprising enough that I would like some clarification from the authors:
  - (*) Bayes Opt can result in performance worse than D(Best in training) on t-RNN Enc -- though it depends on exactly how the final point is selected, this should likely not be possible.
  - "These results indicates that offline optimization of accelerators using Prime can be more data-efficient compared to online methods with active simulation queries" (page 7). This is a very surprising result to me. Online approaches should lead to at least as good accuracy or data efficiency than offline approaches given the  (since any given offline approach can be viewed as an instance of an online approach, whereas the converse is not true). Specifically, I would expect the online version of the fireflies approach to perform better than the offline version (unless there is some characteristic of the surface induced by Prime that is a better fit for optimizing with fireflies). This finding then indicates that either there is significant room for improvement in online approaches for this task, or that the chosen baselines or evaluation metrics are not correct. The paper should include more discussion of this point, or more narrowly scope it to the specific online approaches evaluated in the paper.
- "Top 20% high scoring points in the training dataset are used to provide a validation set..." This is a fascinating point, and I think should be highlighted significantly more in the paper. It seems plausible that this is a key element of the methodology. The conventional wisdom is that DNNs do not extrapolate outside of their training distribution. This approach explicitly requires extrapolation (finding accurate dips between sampled data); because the validation points are not IID and there is a significant amount of hyperparameter search performed, it is possible that selecting for the model that has the best prediction on this set of points selects for a model that is able to generalize to dips in latency space a bit more. I have several questions for the authors based on this point:
  - (*) Is the surrogate model that you ultimately select the exact model resulting from hyperparameter search / validation, or is the model re-trained from scratch with the discovered hyperparameters?
  - Have the authors experimented with other splits between the training and validation set (e.g., training on the best points and validating on the worst)?
  - (*) Is the reported D(Best in training) including the points in this validation set?
  - Are the hyperparameters for the other approaches also tuned using this validation set?
  - (*) What are the actual discovered hyperparameters after this hyperparameter search?
- What fraction of the space is infeasible? How do you do your initial sampling to avoid these infeasible points? What happens if all your final points are infeasible? This is a generally hard problem in high dimensional space. This is a challenge for all optimization algorithms here, the paper doesn't at all discuss the tractability of this sampling or what to do if the final points are all infeasible.
- How sensitive is the optimization to the number of particles, and the parameters of the negative mining optimization?
- As a minor point, alpha is overloaded within the approach: it is used as both the area constraint (Equation 1) and the COM hyperparameter (Equation 2)

[1] Brandon Trabucco, Aviral Kumar, Xinyang Geng, Sergey Levine. Conservative Objective Models for Effective Offline Model-Based Optimization. ICML, 2021.

[2] https://en.wikipedia.org/wiki/Barrier_function

[3] https://en.wikipedia.org/wiki/Multi-armed_bandit#Contextual_bandit

[4] Xin-She Yang. Nature-Inspired Metaheuristic Algorithms. Luniver Press, 2010.


# Response to author response

Thanks to the authors for the very detailed response. All of the responses to the (*) questions are satisfactory, allowing for sufficient understanding of the experimental setting to understand and reproduce results. I'm also pleased to see the additional clarifications and experimental results (especially the clarifications around the hyperparameter search process and the additional validation set experiments) -- these results have convinced me that Prime is doing something beyond just searching for the best point in the validation set. Based on the response, I plan to raise my score to an accept.


**Summary Of The Paper:**

The paper proposes an approach to optimizing parameters of hardware architectures to design architectures that are more efficiently able to execute neural networks. To accomplish this, the paper proposes Prime, an offline optimization algorithm that constructs a surrogate of the design space then optimizes against that design space. Specifically, Prime is trained to be robust to both unseen points in the design space (rather than underestimating their cost) and to infeasible points. For each of these goals, Prima adds a term to the loss function, penalizing adversarial examples and infeasible training points. The authors evaluate Prime by showing that given the same number of evaluations of the ground-truth data, Prime results in lower cost results than online optimization.

**Summary Of The Review:**

The paper proposes an interesting approach to offline design space exploration, and demonstrates that it performs better than online approaches. Though I am not totally satisfied with the evaluation, it is a relatively low lift to bring it up to par with existing literature in the space. A more thorough evaluation would make more a much more compelling paper, but in my view this is not necessary for acceptance.

Weak accept, conditioned on the authors providing more details about the evaluation (specifically, all of the questions marked with (*) above). I would be willing to raise my score further if the authors address some of my other questions (evaluating the sensitivity of the approach to the validation set, discussion the tractability of optimizing with infeasible points, evaluating the amount of compute required to train and evaluate the surrogate, etc).

---

> ### Author Response · Authors · 2021-11-20
> **Author Response (Part 1): (*)-marked questions**
>
> We thank the reviewer for their detailed and thorough feedback and for a positive assessment of our work. We have updated the paper to incorporate answers to various questions (the star-marked questions (*) as well as the other questions) by adding details to Appendices and the main paper, and by providing experiments that investigate various train-test splits. The changes are in $\textcolor{magenta}{magenta}$. We provide the answers below, with corresponding pointers to the paper updates:
>
> ___
> ### (*) marked Questions:
>
> **What is the size of the training set?** We have added the sizes of training sets for different applications in Appendix B.1, last bullet point + Table 17. All of the training set sizes are $\leq 8000$ as mentioned in Section 3. Note that the training datasets are different because different numbers of accelerator designs obtained initially via random sampling may be feasible for different applications.
>
> ___
>
> **Stopping criteria for firefly:** We have now added this detail to **Appendix B.1.1** (paragraph called “Stopping criterion”). In our experiments we ran the Firefly optimizer for atleast 8000 iterations and performed early stopping if the optimal solution found by the optimizer had not changed over the last 1000 iterations. This would typically mean stopping at 8000 iterations, since solutions found by the optimizer plateau after 6000 iterations or so (see convergence curves in **Figure 8**). We also ran online optimization for 4$\times$ as longer (i.e., 32000 iterations) for the t-RNN Enc, t-RNN Dec and MobilenetEdge applications in **Table 18** and found that this made minimal difference on the solution found by the firefly.
>
> ___
>
> **Bayes Opt performance is worse than $\mathcal{D}$(best)** Please note that the dataset $\mathcal{D}$ used in our experiments is collected via random sampling of the accelerator design space, and this dataset is not used by online methods, such as BayesOpt, since they continually query the simulator with design points of their choice. To understand what happens to online optimization methods, we plotted the convergence plot for the evolutionary method (which we found performs generally better than BayesOpt) in **Figure 9**, and we find that these curves very quickly (within 1000 iterations) converge to their final solution for t-RNN Enc and t-RNN Dec. We suspect that this denotes getting stuck in a suboptimal region of the accelerator landscape and being unable to recover, while random-sampling may find a design better than the solution found by these methods.
>
> ___
>
> **Is the surrogate model that you ultimately select the exact model resulting from hyperparameter search / validation, or is the model re-trained from scratch with the discovered hyperparameters?** Yes, we utilize the exact model recovered from hyperparameter search and validation, and do not retrain the model from scratch with the discovered hyperparameters.
>
> ___
>
> **Does the reported D(best) include points in the validation set?** Yes it does, we have added this information in the caption of Tables 3 and 6.
>
> ___
>
> **Actual discovered hyperparameters after search:** We have added these found hyperparameters for $\alpha$, $\beta$ (Equation 3) and checkpoint number for the experiments in Tables 3, 4,  5 and 6 to **Appendix B.1.2, Table 19**.

---

> > ### Author Response · Authors · 2021-11-20
> > **Author Response (Part 2): Additional Questions**
> >
> > ### Additional (non (*)) questions
> >
> > We  answer the additional questions below (we have added some experiments in Appendices as mentioned below), and also applied the suggested text edits in the paper.
> >
> > ___
> >
> > **Total optimization time for PRIME in Figure 5 and 6:** In Figures 5 and 6 the total optimization time simply corresponds to the wall-clock time needed to run a forward pass on the conservative neural network surrogate trained by PRIME on a CPU, and therefore it increases with more optimization iterations, since the firefly optimizer queries the learned surrogate model in each iteration. We have clarified this detail in the caption of Figure 5.
> >
> > ___
> >
> > **Other splits between train and validation set:** To answer this question, we have now run experiments training PRIME on other splits of the training and validation set. In particular, we evaluated two other train-val splits on three applications corresponding to Table 3. We have added these to a new **Appendix A.9, Table 16** in the paper. We find that utilizing the top fraction of data can help in certain cases, but it can also reduce sample size for cross-validation in certain small-dataset applications, leading to worse performance.
> >
> > ___
> >
> > **Are the hyperparameters for other methods tuned using the validation set?** For the online methods we study, we directly utilize the online performance evaluated against the simulator for cross-validation and choosing hyperparameters, as it is the groundtruth performance metric that we wish to optimize. We have now added the details of tuning the Firefly optimizer in **Appendix B.1.1.** (paragraph titled “Hyperparameter tuning for firefly”). Briefly, we adaptively adjust hyperparameters for firefly over the course of online optimization, which gives us a favorable balance between time and compute requirements for running firefly against the simulator and tuning hyperparameters. For COMs baseline in Appendix A, which is offline, we use the validation set + Kendall’s rank correlation metric as well for tuning.
> >
> > ___
> >
> > **Scoping the discussion around online approaches studied in this paper:** We agree with the reviewer that, in general, there always exists an online method that would perform better than the offline method (e.g., an online method that incorporates PRIME itself). We would like to clarify that the goal of this paper is not to advocate for replacing online methods by offline methods, but rather to show that offline methods can perform well and can enable data reuse, with the point that these methods should be used in practice for the accelerator design problem. We have edited the discussion section (**Section 7**) to incorporate this point.
> >
> > Additionally, as we also find in Table 8, this evolutionary approach that we utilized outperforms recently published state-of-the-art online optimization methods (such as [P3BO](https://arxiv.org/abs/2006.03227)). As a result we used the evoltuionary algorithm as the main point for comparison.
> >
> > ___
> >
> > **Sensitivity of negative mining to number of negative particles:** Thank you for the great question! We are running this ablation on some domains, and will get back to you.
> >
> > ___
> >
> > **What fraction of the space is infeasible? How do you prevent initial sampling to avoid these infeasible points?** We have added a discussion of this point in **Appendix  B.1, last bullet point** (Page 23, “Discussion on data quality”). Exhaustive enumeration of the entire search space is intractable due to its large number of data points (more than 452M). Among our randomly sampled accelerator designs, on an average, ~71% of the data are infeasible. We agree that random sampling may only find points that are infeasible, which will not be sufficient to train our conservative surrogate, however, we used this scheme only for simplicity reasons. In such scenarios, more sophisticated schemes such as utilizing data from previous runs of an online optimization algorithm mixed with data collected from random sampling may be a better strategy. We have added this discussion in Appendix B.1, last bullet point.
> >
> > ___
> >
> >
> > **Total compute for PRIME:** We train one run of the conservative surrogate in PRIME on 1 P100 GPU, and it requires about 2 hours for 10k gradient steps for the runs in Table 3. We believe that this is not the most efficient implementation of PRIME, as it is multi-threaded and requires calling subprocesses for the negative sampling part, and can be improved. We typically train our surrogates for 200k gradient steps, which means that training PRIME needs about 40 hours in our setup. We believe this is slow just because of our implementation, and the total time needed to train PRIME can be reduced significantly by using GPU parallelism, or advanced accelerators such as TPUs.

---

### Official Review · Reviewer_7gpC · 2021-11-03

**Correctness:** 3
**Technical Novelty And Significance:** 3
**Empirical Novelty And Significance:** 3
**Recommendation:** 6
**Confidence:** 3

**Main Review:**

Strength:
- This work attacks a very commonly seen problem from prior work: the over-optimism and the uselessness of the infeasible design points in hardware performance modeling. Problems are well-defined and solutions are targeted and justified. The technical presentation of the methods proposed is solid.
- Experiments are extensive and have compared offline and online methods very thoroughly. The paper also shows the results of applying PRIME on other accelerator designs to examine the applicability.
- Writing is clear and the paper is easy to follow.

Weakness:
- The challenge of simulation time in online methods is now shifted to dataset construction. How is the cost of constructing the training datasets, especially when a new dataset is required for a new context (application)?
- Figure 12 shows the challenge of overoptimism in performance modeling. However, there is no experiment showing the difference of predicted and measured latency/energy when using the proposed method. It is not clear whether the results of PRIME are produced by the performance model or by the simulation/measurement using the optimization results.
- The results in Table 5 are not enough to show the capability of PRIME under the zero-shot setting. One of the benefits of the offline method is to reuse the logged training data. It would be better if authors could show the results on all benchmarks (instead of 1 or 2 networks) when using different training application subsets. Meanwhile, given a new application, when will PRIME choose to build a new training dataset and when will PRIME choose to use a zero-shot setting? How should PRIME choose the training applications? What is the minimal training application subset to get the best results on all benchmarks?


**Summary Of The Paper:**

This paper mainly focuses on the architectural sizing of the neural network accelerators. Optimizing accelerator parameters is expensive due to the time-consuming simulation and numerous infeasible design candidates. This work presents a data-driven offline optimization framework, PRIME, that reuses the previous simulation results to produce new designs without additional active queries to an explicit silicon or simulator. The key idea is to train a conservative surrogate model, that predicts the cost (e.g. latency) of an architecture candidate, by minimizing the prediction error of feasible designs and penalizing the infeasible designs for avoiding optimism. Experiments show that PRIME offers 1.20x - 1.54x speedup on average over manual design and 1.26x speedup for unseen applications.

**Summary Of The Review:**

This work presents an effective framework PRIME to tackle the challenges of automating hardware design optimization. Though it seems to be an incremental medication of previously published efforts, the proposed method is clear and useful. The paper could be stronger with a better analysis as stated in the Main Review.

---

> ### Author Response · Authors · 2021-11-20
> **Author Response**
>
> We thank the reviewer for their feedback and for a positive assessment of the work. We have revised the paper to address the reviewer’s comments to add the overestimation curve (**Figure 14**) and zero-shot results (**Appendix A.8**) and we summarize the responses to the answers below.
>
> ___
>
> **Cost of collecting the dataset:** When only optimizing for a single-application, we need to collect a database of designs from the simulator that would incur a cost equivalent to online methods (note however, that PRIME can often achieve better performance than online methods in the same data budget as shown in Table 5). However, this **one-time** collected dataset can be *reused* by PRIME to optimize:
> - accelerators for unseen applications without collecting new data (zero-shot setting, **Table 5**)
> - accelerators for different combinations of applications (multi-application setting, **Table 4**).
> - accelerators under variable design constraints (**Appendix A.2, Table 10**).
>
> In contrast, an online method would require consistent access to the simulator as design constraints change, or when a new application is added or if we need to optimize accelerators for a different combination of applications. Therefore, the cost of data collection is amortized over various applications and designs: while PRIME does need O(N) datapoints, these data-points can be be reused under many scenarios discussed above, and we show examples of such scenarios in the paper (Tables 4, 5), online methods need to collect O(N * K) data points, where K is the number of optimizer scenarios we consider, and K can be very large when target applications or design constraints can change.
>
> ___
>
> **Overestimation plot for PRIME:** We have now added a scatter plot demonstrating the overestimation in PRIME in **Figure 15, Page 29**. Comparing the plot to Figure 14 in the paper which shows the overestimation for a naive surrogate model, we observe that PRIME predicts a higher latency for a number of models compared to their actual latency -- meaning that PRIME **does not** assign low latencies to designs with high ground truth latencies. In fact it predicts higher latencies for most points.
>
> ___
>
> **Zero-shot results on all the benchmarks; minimal subset of applications:** For the conservative surrogates trained by PRIME in Table 4, we are evaluating them on all the applications we study. A subset of the results which are complete are now shown in **Appendix A.8., Table 15, Page 20**, and depict that PRIME can outperform online evolutionary methods even when evaluated over all applications in a zero-shot manner. Additionally, we experimented with a few more subsets of training applications, with the goal to identify the minimal subset of applications that gives rise to good results on all benchmarks. We are running these experiments and will update the reviewer and the paper before the end of the rebuttal period.
>
> ___
>
> **It is unclear if the results of PRIME are produced by the performance model or by simulation/measurement using the optimization results:**  Per our interpretation of the question, we would like to clarify that the results we report in Tables 3, 4, 5 and 6 in the main paper correspond to the actual latency of the optimized designs found by PRIME in a purely offline fashion, which are then evaluated under the simulator to produce actual latencies. More details of our evaluation protocol can be found in the paragraph titled  "Evaluation Protocol" in Section 3. We are happy to clarify any details if something is unclear.
>
> ___
>
> **Given a new application, when should we collect a new dataset, when will it be used in a zero-shot setting.** This is a great question! In general, this kind of a question is hard to precisely answer for learned neural network models, even in well-studied problems such as image classification. For the problem of hardware accelerator design, we suspect that a heuristic for answering this question could be by comparing the ratio of compute required to the application size. For applications with drastically different compute to size ratio compared to what is seen in the training dataset, we expect that collecting a dataset might be more helpful, whereas if the test application admits a similar value of this ratio as an already existing application, we might be able to utilize the zero-shot generalization.

---

> > ### Author Response · Authors · 2021-11-23
> > **Author Response (Part 2)**
> >
> > Dear Reviewer 7gpC,
> >
> > We have now updated the paper to add an ablation study in a new **Appendix D** (changes in $\textcolor{magenta}{magenta}$) aimed at identifying the minimal subset of training applications that still allow us to generalize to all the nine applications considered in the paper. While exhaustively searching all subsets of training applications is computationally intractable, we were able to find a set of **four** applications (MobileNetV2, MobileNetV3, M5, M6) that still allow us to attain good zero-shot performance on all the nine applications considered in the paper (**Table 20** presents the results).
> >
> > We have also added a discussion of guidelines that may allow a practitioner to choose between zero-shot generalization of PRIME and collecting new data for a new target application. While this is hard to precisely quantify (for any machine learning method, even in well-studied domains such as image classification), based on our experiments, we found that PRIME could generalize to applications with similar compute/memory requirements, and therefore four applications were enough for the applications we considered. Therefore, we suggest that a reasonable practical heuristic could be to evaluate if the target test applications admit similar compute/memory statistics as the training set of applications or not. If the compute/memory statistics are similar, we might be able to utilize the zero-shot generalization properties of PRIME. If not, we could collect training data for the new application. We have added a discussion of this in Appendix D (under paragraph titled "Practitioner guidance").
> >
> > Thanks!

---

### Official Review · Reviewer_TEzV · 2021-11-08

**Correctness:** 3
**Technical Novelty And Significance:** 2
**Empirical Novelty And Significance:** 3
**Recommendation:** 6
**Confidence:** 3

**Main Review:**

The strengths of the paper are as follows:
+ This paper provides a new perspective for tackling the overfitting of hardware efficiency predictors by leveraging adversarial training and infeasible design samples;
+ The developed tool can be useful as an early-stage development tool, and the released accelerator dataset can be useful to the community;
+ This paper is the first to consider unseen applications in a zero-shot setting when it comes to accelerator optimization.

The weaknesses of the paper are below:
- This paper does not review, discuss, and evaluate against another group of related methods for designing hardware accelerators, which considers automated accelerator designs, e.g., [MAGNet, ICCAD 2019] and AutoDNNchip, FPGA 2020];
- This work does not evaluate the resulting accelerators with any SOTA dedicated accelerators. I think it is not convincing for a accelerator tool to be practical without evaluating where the resulting design with the dedicated expert design for the same application/task.

**Summary Of The Paper:**

The authors first introduce two practices for application-specific hardware accelerators: 1) Designers need to spend considerable manual effort and perform large number of time-consuming simulations to find accelerators that can accelerate multiple target applications while obeying design constraints. Moreover, such a simulation-driven approach must be re-run from scratch every time the set of target applications or design constraints change. 2) An alternative paradigm is to use a data-driven, offline approach that utilizes logged simulation data, to architect hardware accelerators, without needing any form of simulations. Such an approach not only alleviates the need to run time-consuming simulation, but also enables data reuse and applies even when set of target applications changes.

As such, they develop such a data-driven offline optimization method for designing hardware accelerators, dubbed PRIME, which learns a conservative estimate of the desired cost function, utilizes infeasible points, and optimizes the design against this estimate without any additional simulator queries during optimization. The authors evaluate PRIME architects accelerators---tailored towards both single- and multi-applications as well as unseen applications in a zero-shot setting.

**Summary Of The Review:**

This paper develops a data-driven offline optimization for identifying good hardware accelerator designs, and conducts sufficient ablation studies to validate their methods, while the evaluation against SOTA methods can be improved.

---

> ### Author Response · Authors · 2021-11-18
> **Author Response**
>
> We thank the reviewer for their feedback and positive assessment of our work. To address the reviewer’s comments, we have revised the paper (changes in $\textcolor{magenta}{magenta}$). We summarize the changes below, including answers to the reviewer’s questions:
>
> 1. Thank you for pointing us to very related prior works. We have now discussed these related works (MAGNet, AutoDNNchip) in our Related Work Section 5.
>
> 2. We have now added comparisons between the accelerators found by PRIME and the [EdgeTPU accelerator](https://arxiv.org/abs/2102.10423), which is a prior accelerator designed by human experts, aimed towards accelerating image-classification tasks in **Appendix A.7**. As shown in **Table 14**, across all the nine applications we study in this paper, we find that PRIME finds accelerators that are **2.69$\times$** (up to 11.84$\times$ in t-RNN Enc) better than the EdgeTPU accelerator in terms of latency. PRIME attains this improvement on average, reducing the chip area usage by 1.50$\times$ (up to 2.28$\times$ in MobileNetV3). Even when evaluated only on the MobileNet image-classification domains, we attain an average improvement of **1.85$\times$** in terms of latency. This indicates that PRIME is effective when compared to the human-engineered EdgeTPU accelerator.
>
> Please let us know if this experiment addresses your questions.

---

### Decision · Program_Chairs · 2022-01-20

**Decision:**

Accept (Poster)

**Comment:**

The authors give an effective framework PRIME to tackle the challenges of automating hardware design optimization.  This problem is of importance to the community.  Overall, the reviewers thought the paper gave a nice clean approach to the problem and that the community would be interested with these results.